# Gradient Explosion and Representation Shrinkage in Infinite Networks

## Abstract

We study deep fully-connected neural networks with layer normalization using the mean field formalism, and carry out a non-perturbative analysis of signal propagation. As a result, we demonstrate that increasing the depth leads to gradient explosion or to another undesirable phenomenon we call representation shrinkage. The appearance of at least one of these problems is not restricted to a specific initialization scheme or a choice of activation function, but rather is an inherent property of the fully-connected architecture itself. Additionally, we show that many popular normalization techniques fail to mitigate these problems. Our method can also be applied to residual networks to guide the choice of initialization variances.

## 1 Introduction

Deep learning is arguably the most successful modern machine learning method when it comes to modelling complex data. Informally, this is often attributed to hidden representations not being constrained by the designer's knowledge. Empirically, deep neural networks tend to outperform shallow ones, which is explained by them learning a richer hierarchy of representations. There is a body of works making precise a number of aspects of this idea, e.g. showing that expressivity (Montufar et al., 2014) and disentangling ability (Poole et al., 2016) grow with the network's depth.

Because large depth is often desired, much effort was devoted to networks with many layers. There are three main groups of techniques to aid their training: normalization, critical initialization and skip connections. Normalizations ensure the right scale of certain preactivation statistics, for example magnitude or batch moments. Critical initialization is based on mean-field analysis of signal propagation. It aims at picking an initialization scheme that brings the singular values of the input-output Jacobian close to 1. Skip connections are used in residual blocks, which learn the difference between the target function and the identity. Such construction guarantees that the network can represent the identity. A different approach with similar motivation is to use a parameterized activation functions, which are linear for some value of its parameters, so that not only the identity but every linear map is guaranteed to be representable. Our results demonstrate a limitation of normalization methods and critical initialization, and naturally suggest initializing deep networks as linear maps.

A different line of research studied the physics of large networks. It is known that wide but shallow neural networks at initialization simplify to Gaussian processes (GP) (Neal, 1996), and their training is shown to be equivalent to kernel methods (Jacot et al., 2018). However, realistic finite networks outperform their infinite limits (Ghorbani et al., 2020), which is explained by the lack of feature learning in large networks (Chizat et al., 2019). The situation with deep networks is qualitatively different – the GP regime breaks down when depth and width are large but comparable. Therefore, a well-behaved infinite depth and width limit would provide a promising starting point for a theoretical model capable of capturing feature learning. Suitably scaled infinite residual networks (ResNets) become diffusion processes; one could hope that by appropriately combining different activation functions and normalization methods, one could obtain a similar behaviour in MLPs. We hope that by explaining why this is impossible, our result will help to illuminate most promising future directions in physics of large networks.

The main advice of our work – that there are limitations to increasing the depth, even at critical initialization – was also noted in Pennington et al. (2017) and Pennington et al. (2018). This line of work expresses the input-output Jacobian as a product of random matrices, and uses random matrix

theory (RMT) to analyse its full spectrum. They find that the maximum and variance of its singular values grow with depth even if the mean remains $O(1)$.

## 1.1 RELATED WORK

**Normalizations:** Rescaling the preactivations to zero mean and unit variance across a mini-batch was introduced in Ioffe & Szegedy (2015); across all the neurons in single layer was done explicitly in (Lei Ba et al., 2016) and implicitly in (Klambauer et al., 2017). For an overview see Huang et al. (2020). Mean-field analysis of batch normalization was carried out in Yang et al. (2019).

**Critical initialization:** Mean-field formalism, describing signal propagation in wide networks, was introduced in Poole et al. (2016) based on ideas from Neal (1996). It was applied in Schoenholz et al. (2017) to characterize Lyapunov exponents for signal and gradient propagation in fully-connected networks. The impact of activation function on initialization was investigated in Hayou et al. (2019); this work, as well as Xiao et al. (2019), note polynomially-quick convergence of correlation with depth at criticality. A form of correlation degeneracy was noted in Daniely et al. (2017). Initialization of convolutional networks was discussed in Xiao et al. (2018).

**Skip connections:** Residual blocks were introduced in He et al. (2016). Decreasing the variance of residual weights with depth was shown to be necessary for a non-trivial kernel (Hayou et al., 2021) and good signal propagation with a class of activations (Yang & Schoenholz, 2017); its utility was observed empirically in (Wang et al., 2018). Trainable activations, brought closer to linear at the initialization, were reported to help in training very deep networks in He et al. (2015).

**Limiting behaviour of large nets:** Breakdown of GP regime was demonstrated in Hanin & Nica (2020). Physics of large networks was studied systematically in Roberts et al. (2021). The correspondence between infinitely deep ResNets and diffusion processes was described in Peluchetti & Favaro (2020), and generalized to doubly infinite ResNets in Peluchetti & Favaro (2021). A different kind of problems with wide and deep ReLU networks was analysed in Hanin & Rolnick (2018).

## 1.2 CONTRIBUTION

To the best of our knowledge, we present the first general non-perturbative result on signal propagation in infinitely wide networks. It holds for arbitrary[1] activation functions, and they can vary between layers. Our proof does not rely on the assumption of independence between gradients and weights. On a high level, our framework explains the origin of "signal distortion" in fully-connected networks, and explains *why* they are harder to stack than residual networks. Unlike RMT, our method can handle pairs of data points with non-infinitesimal differences.

We provide a precise trade-off between having rich hidden representations and well-behaved gradients in fully-connected and residual networks, making rigorous the intuition of competition between these two goals. Unlike in previous work, our result is non-asymptotic in depth and avoids formulation "up to a constant". We hope that improvements to our result will serve as a guidance for architectural choices.

## 2 SET-UP

## 2.1 NOTATION

We denote the inner product and vector norms of $u, v \in \mathbb{R}^d$ as

$$u.v = \sum_{i=1}^{d} u_i v_i \qquad \|u\| = \sqrt{u.u}$$

For a matrix $M \in \mathbb{R}^{n \times m}$, it will be natural to induce the operator norm by the root-mean-squared norm

$$\|M\|_{op} = \max_{v \in \mathbb{R}^m} \frac{n^{-\frac{1}{2}} \|Mv\|}{m^{-\frac{1}{2}} \|v\|}$$

---

[1]Technically activations have to be square-integrable with respect to Gaussian weight

Gaussian covariance will be denoted as

$$\mathfrak{C}[f|\rho] = \mathbb{E}\left[f(x)f(y)\middle| \begin{pmatrix} x \\ y \end{pmatrix} \sim \mathcal{N}\left(0, \begin{pmatrix} 1 & \rho \\ \rho & 1 \end{pmatrix}\right)\right].$$

To compare activation functions we will be using the Gaussian norm

$$\|\phi\|_{\mathcal{N}}^2 = \mathbb{E}\left[\phi\big(\mathcal{N}(0,1)\big)^2\right] = \mathfrak{C}[\phi|1]$$

We linearize functions by truncating their Hermite expansion

$$\overline{\phi}(x) = \hat{\phi}_0 + \hat{\phi}_1 x \qquad \text{for} \qquad \phi = \sum_{k=0}^{\infty} \hat{\phi}_k h_k$$

where $h_k$ are normalized Hermite polynomials[2]. Let us note that they satisfy $\mathbb{E}[h_k(x_1)h_{k'}(x_2)] = \delta_{kk'}\rho^k$ for jointly normal zero-mean $x_1, x_2$ of variances 1 and covariance $\rho$ (for a proof, see for example O'Donnell (2021)).

## 2.2 SET-UP

We will focus on an untrained fully-connected neural network of depth $L$ and $N_l$ neurons in layer $l$. We compose it from pointwise non-linearities, affine transforms and layer normalizations[3]; formally the output of the $i$-th neuron in layer $l$ on data point $x_\alpha$ is given by

$$z_i^{(l)}(x_\alpha) = \frac{\sqrt{N_l} \cdot y_i^{(l)}(x_\alpha)}{\|y^{(l)}(x_\alpha)\|} \qquad \text{where} \qquad y^{(l)}(x_\alpha) = \frac{1}{\sqrt{N_{l-1}}} \sum_{j=1}^{N_{l-1}} W_{ij}^{(l)} \phi_l\big(z_j^{(l-1)}(x_\alpha)\big) + b_i^{(l)} \quad (2.1)$$

where weights $W_{ij}^{(l)} \in \mathbb{R}^{N_l \times N_{l-1}}$ and biases $b_i^{(l)} \in \mathbb{R}^{N_l}$ have variances $\sigma_{(l),w}^2, \sigma_{(l),b}^2$ respectively, and $\phi_l : \mathbb{R} \circlearrowleft$ is the activation function of $l$-th layer. We will refer to $z^{(l)}$ as *preactivations*. At the first layer, we set $\phi_1(x) = x$ and regard it as an "embedding".

An important role in our analysis will be played by *correlation* or cosine similarity of the preactivations, defined for two input data points $x_\alpha, x_\beta$ as

$$\rho_l = \frac{1}{N_l} \sum_{i=1}^{N_l} z_i^{(l)}(x_\alpha) z_i^{(l)}(x_\beta) = \frac{y^{(l)}(x_\alpha) \cdot y^{(l)}(x_\beta)}{\|y^{(l)}(x_\alpha)\| \cdot \|y^{(l)}(x_\beta)\|} \qquad (2.2)$$

When the widths are very large $N_l \to \infty$, with $x_\alpha$ fixed and the parameters $(W_{ij}^{(l)}, b_i^{(l)})$ having independent entries[4] (e.g. distributed according to the Gaussian initialization scheme), one can use the mean field approximation (Poole et al., 2016) and treat $z_i^{(l)}(x_\alpha)$ as a normal random variable. Then, by the law of large numbers applied to equation 2.2, the correlation changes as

$$\rho_l = P_l(\rho_{l-1}) \overset{\text{def}}{=} \frac{\sigma_{(l),b}^2 + \sigma_{(l),w}^2 \mathfrak{C}[\phi|\rho_{l-1}]}{\sigma_{(l),b}^2 + \sigma_{(l),w}^2 \mathfrak{C}[\phi|1]} \qquad (2.3)$$

Letting $\phi = \sum_{k=0}^{\infty} \hat{\phi}_k h_k$ be the expansion of the activation function $\phi$ into normalized Hermite polynomials, we can work out the Gaussian covariance and obtain

$$P(\rho) = \frac{\sigma_b^2 + \sigma_w^2 \hat{\phi}_0^2}{\sigma_b^2 + \sigma_w^2 \|\phi\|_{\mathcal{N}}^2} + \frac{\sigma_w^2}{\sigma_b^2 + \sigma_w^2 \|\phi\|_{\mathcal{N}}^2} \sum_{k=1}^{\infty} \hat{\phi}_k^2 \rho^k \qquad (2.4)$$

**Observation 1.** *From expression 2.4 it is apparent that $P$ is a power series with non-negative coefficients. Therefore, it is non-decreasing and convex on $[0,1]$. This implies existence and uniqueness of an attracting (stable) fixed point [5] of $P$.*

---

[2]First two are $h_0(x) = 1$, $h_1(x) = x$

[3]Without centering

[4]Technically, we also need regularity conditions (e.g. finite fourth moment) to ensure that CLT applies. These hold in most practical scenarios.

[5]A solution to $P(\rho_{\text{fp}}) = \rho_{\text{fp}}$, such that $P^n(\rho) \overset{n \to \infty}{\longrightarrow} \rho_{\text{fp}}$ for all $\rho$ in some neighbourhood of $\rho_{\text{fp}}$. A sufficient condition for the latter is $P'(\rho_{\text{fp}}) < 1$.

This is relevant when $\sigma_w, \sigma_b, \phi$ are the same across layers; then forward propagation affects the correlation by repeated application of $P$, and $\rho_l$ approaches $\rho_{\text{fp}}$ as $l$ grows. For $\rho_0 \approx 1$ we have $1 - \rho_l \approx P'(1)^l(1 - \rho_0)$, so the derivative of $P$ at 1 determines whether the fixed point $\rho = 1$ is stable or unstable. Depending on the ratio $\frac{\sigma_b}{\sigma_w}$ we have three regimes of initialization, analogous to the phase diagram from Schoenholz et al. (2017):

- Ordered initialization for $P'(1) < 1$. Then $\rho_{\text{fp}} = 1$ is stable and $\rho_l \to 1$ exponentially quickly with $l$. Preactivation perturbations decay, so hidden representations of all data become strongly aligned and gradients vanish.

- Critical initialization for $P'(1) = 1$. Then $\rho_{\text{fp}} = 1$ is stable and $\rho_l \to 1$ polynomially quickly with $l$. Norm of preactivation perturbations changes sub-exponentially quickly so the gradients are well-behaved.

- Chaotic initialization for $P'(1) > 1$. Then, the fixed point $\rho = 1$ of $P$ is repelling (unstable) and there exists another fixed point $\rho_{\text{fp}} < 1$ which is attracting. In that case $P'(\rho_{\text{fp}}) < 1$, so $\rho_l \to \rho_{\text{fp}}$ exponentially quickly with $l$. Hidden representations of similar data points are pulled far apart, preactivation perturbations grow and gradients explode.

## 3 MAIN RESULT

Our main result, theorem 1, quantifies the trade-off between steepness, representation ampleness, non-linearity and depth.

**Theorem 1.** *Let $z^{(L)}$ be a wide fully-connected network as described in equation 2.1: having $L$ layers, using layer normalization and employing activation function $\phi_l$ at layer $l$. In the limit $n_l \to \infty$ we have the following inequality*

$$\left\|\frac{\partial z^{(L)}}{\partial z^{(0)}}\right\|_{op}^2 \geq \frac{(1-\rho_{\max})^2}{8} \sum_{l=1}^{L} \frac{\left\|\phi_l - \overline{\phi}_l\right\|_{\mathcal{N}}^2}{\|\phi_l'\|_{\mathcal{N}}^2}$$

*where $\frac{\partial z^{(L)}}{\partial z^{(0)}}$ is the input-output Jacobian, $\left\|\phi - \overline{\phi}\right\|_{\mathcal{N}}^2$ is the error of linearizing the activation function $\phi$ and the maximal correlation $\rho_{\max}$ is defined as*

$$\rho_{\max} = \max_{1 \leq l \leq L} \min_{x_\alpha, x_\beta} \frac{z^{(l)}(x_\alpha) \cdot z^{(l)}(x_\beta)}{N_l}$$

We delay the complete proof to appendix A, and in this section just sketch the main ideas. Let us start with the consequences of theorem 1, and give interpretations for each quantity. Theorem 1 tells us that at least one of the following undesirable effects will necessarily occur:

1. The Jacobian norm $\left\|\frac{\partial z^{(L)}}{\partial z^{(0)}}\right\|_{op}$, which measures the steepness of $z^{(L)}$ as a function of $z^{(0)}$, is large. This means that a small change in the input drastically alters the output and the network becomes overly sensitive to perturbations of data. In Bayesian interpretation, the prior hypothesis favours jagged functions. Then the gradients with respect to the first layer parameters are necessarily large, which means gradient explosion. This happens when $\frac{\sigma_{(l),b}}{\sigma_{(l),w}}$ are predominantly small, resembling the chaotic phase.

2. The maximal correlation is close to one $\rho_{\max} \approx 1$. This means that at some layer $l$, the hidden representation $z^{(l)}(x)$ of any possible input data point $x$ lies in the cone $\frac{v \cdot z^{(l)}(x)}{N_l} > \rho_{\max}$ for some vector $v$. We call this scenario *representation shrinkage*. It may lead to a range of problems. First, the representations do not utilize the whole available space but rather are confined to a small region. Second, a rich representation may be necessary for some problems. In this case getting rid of the shrinkage may be a prerequisite for learning. However, it costs time. We have observed that the alignment of preactivations tends to disappear layer-by-layer rather than at all the layers simultaneously, suggesting that deeper networks require more optimization steps. Third, if the angle between preactivations becomes comparable to the machine precision, then in practice gradient descent may run into numerical

issues. Fourth, suppose that $z^{(l)}(x) \approx v$ for any data point $x$. If the following block $z^{(l+k)} \circ \cdots \circ z^{(l+1)}$ of $k$ layers is reasonably smooth, then for any input it ever encounters, the block will we well approximated by its linearization at $v$. Therefore, $z^{(l+k)} \circ \cdots \circ z^{(l+1)}$ does little processing, but requires the full computational budget to implement. This scenario happens when $\frac{\sigma_{(l),b}}{\sigma_{(l),w}}$ tend to be large (ordered-phase-like behaviour).

3. The linearization error $\|\phi_l - \overline{\phi_l}\|_{\mathcal{N}}$ is small. This restricts the choice of activation functions we can use to ones that are close to linear. The layers need to be close to linear maps, so informally they will not "process the data too much". This forces us to make the MLP into something resembling a ResNet. At zero error the network is completely linear, so forward propagation perfectly preserves norms, covariances and correlations of preactivations. With no correlation distortion the gradient-shrinkage trade-off disappears, which is reflected by theorem 1 becoming vacuous.

4. The depth $L$ upper-bounded. Sum of many lower-bounded terms cannot stay upper-bounded, so we cannot stack as many layers as we want.

*Sketch of proof.* We examine the cosine similarity of output preactivations $\rho_L$ as a function of cosine similarity of inputs $\rho_0$. This functional relationship in an example network is illustrated on figure 1. For infinitely wide networks, this functional dependence is exactly described by

$$\rho_L = P_L \circ \cdots \circ P_1(\rho_0)$$

where $P_l$ are taken from equation 2.3. By observation 1, each $P_l$ is convex. By adding layers we compose more and more $P$'s, and we can show that the convexity "accumulates". Then, the "flatness near 0" and the "steepness near 1" become more pronounced and the transition between them becomes sharper[6]. Therefore, the infimum[7] of $P_L \circ \cdots \circ P_1$ is close to 1, or its derivative at 1 is large. In the former case every possible pair of output preactivations is strongly aligned and $\rho_{\max} \approx 1$; in the latter the Jacobian norm and thus gradients are large.

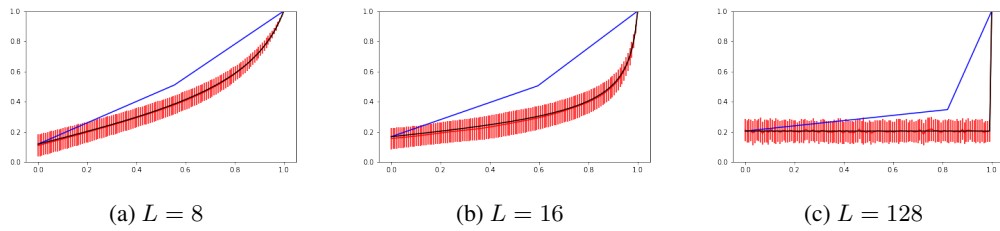

(a) $L = 8$        (b) $L = 16$        (c) $L = 128$

Figure 1: Output correlation $\rho_L$ as a function of input correlation $\rho_0$, for the erf activation. Empirical ($N_l = 1024$) in red, infinite-width in black. Blue is an upper bound on $P_L \circ \cdots \circ P_1$ – for details see remark 2 in appendix A.1.

The underlying phenomenon is the pathological behaviour of $\rho_L$ as a function of $\rho_0$. Every $z^{(L)}$ has correlation 1 with itself, so $\rho_L(\rho_0 = 1) = 1$. However, we can find an upper-bound (blue on figure 1) that tends to $P_L \circ \cdots \circ P_1(0)$ pointwise on $[0, 1)$ as $L$ grows. $\square$

**Remark 1.** *One might ask where does gradient vanishing fit in this framework. In fact, it is a special case of representation shrinkage. This is best illustrated on the simplified example with $\sigma_{(l),W}, \sigma_{(l),b}, \phi_l$ independent of $l$. Gradient vanishing happens at the ordered initialization, when representation shrinking "proceeds exponentially quickly" with the number of layers, i.e. $1 - \rho_l = O(e^{-\lambda l})$. One way to think about theorem 1 is that criticality is not sufficient to have a well-behaved network: even though it avoids gradient explosion and vanishing, representation shrinkage still happens "at a polynomial rate" (theorem 1 allows to deduce the rate at least $1 - \rho_l = O(l^{-\frac{1}{2}})$).*

Now let us apply our technique to residual networks. We consider residual blocks consisting of an affine transform, normalization, pointwise activation and a linear map. After adding the residual we normalize the signal. Formally, it is described by the equations

---

[6]Pictorially, this resembles the behaviour of the graph of the function $y = a + (1 - a)x^\alpha$ as $\alpha$ gets large. When $a \approx 1$ these functions have infimum close to 1, otherwise their slope (derivative) at 1 is big.

[7]By observation 1, the infimum on $[0, 1]$ is attained at $\rho = 0$.

$$Y^{(l)} = U^{(l)} z^{(l-1)}$$

$$y^{(l)} = z^{(l-1)} + \frac{1}{\sqrt{m_l}} V^{(l)} \phi_l \left( \frac{\sqrt{m_l}}{\|Y^{(l)}\|} Y^{(l)} \right)$$

$$z^{(l)} = \frac{\sqrt{n}}{\|Z^{(l)}\|} Z^{(l)}$$

Where $z^{(0)} = x \in \mathbb{R}^n$ is the input. The parameters are $U^{(l)} \in \mathbb{R}^{m_l \times n}$, $V^{(l)} \in \mathbb{R}^{n \times m_l}$; we assume their entries are independent with variances $1$ and $\sigma_{(l)}^2$ respectively.

**Theorem 2.** *For such residual network in the limit $n, m_l \to \infty$ we have*

$$\left\| \frac{\partial z^{(L)}}{\partial z^{(0)}} \right\|_{op}^2 \geq \frac{(1-\rho_{\max})^2}{8} \sum_{l=1}^{L} \frac{\sigma_{(l)}^2 \|\phi_l - \overline{\phi_l}\|_{\mathcal{N}}^2}{1+\sigma_{(l)}^2 \|\phi_l'\|_{\mathcal{N}}^2}$$

*where as before*

$$\rho_{\max} = \max_{1 \leq l \leq L} \min_{x_\alpha, x_\beta} \frac{z^{(l)}(x_\alpha) \cdot z^{(l)}(x_\beta)}{N_l}$$

In addition to the four undesired phenomena of the fully-connected variant 1, theorem 2 about ResNets permits another alternative: initialization variances $\sigma_{(l)}$ of weights $V^{(l)}$ being small. This situation is not problematic; in the extreme case $\sigma_{(l)} = 0$ the network is initialized as the identity.

## 4 EXPERIMENTS

### 4.1 APPLICATIONS OF THEOREM 1

We show the trade-off between non-linearity and representation shrinkage on the example of critically initialized leaky ReLU networks. For a leaky ReLU network with negative slope $a$ and $\sigma_b = 0$, the inequality 1 predicts

$$1 - \rho_{\max} \leq \sqrt{\frac{16\pi}{\pi-2}} \cdot \frac{1}{\sqrt{L}} \cdot \frac{\sqrt{1+a^2}}{1-a} \tag{4.1}$$

For detailed derivation of the constants see appendix B. We see that adding layers with constant $a < 1$ (figure 2a) and decreasing $a$ at constant depth (figure 2b) decrease $1 - \rho_{\max}$, i.e. cause representation shrinkage at some layer. To avoid this effect, we either need to make $L$ bounded and the network shallow or $a \approx 1$ and keep $\text{LReLU}_a$ close to linear.

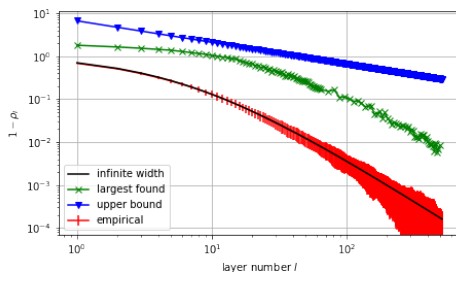

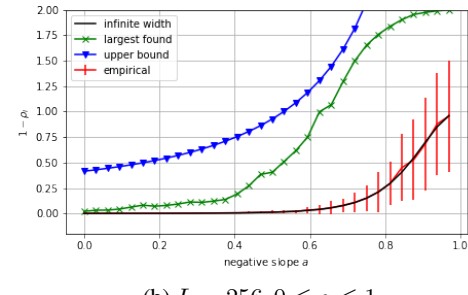

(a) $a = 0$ (standard ReLU), $1 \leq L \leq 512$

(b) $L = 256, 0 \leq a \leq 1$

Figure 2: $1 - \rho_L$ as function of depth $L$ (fig 2a) and negative slope $a$ (fig 2b). Empirical correlations for sample data pairs in red; infinite-width predictions in black; smallest correlation (largest $1 - \rho_l$) found by gradient descent on data (with network parameters fixed) in green; bounds obtained by rearranging inequality 4.1 in blue. Lower means smaller $1 - \rho$, i.e. stronger shrinkage.

We illustrate the relationships between gradient explosion and representation shrinkage on the example of erf[8] activation. With critical initialization, the operator norm of the network Jacobian (3c)

---

[8]Smooth increasing function $\mathbb{R} \twoheadrightarrow [-1, 1]$ defined as $\text{erf}(x) = \frac{2}{\sqrt{\pi}} \int_0^x e^{-y^2} dy$

does not grow with depth, and thus gradients (3e) are well-behaved; however, the whole input space gets progressively more squeezed at deeper layers (3a). Theorem 1 predicts that this happens at the rate at least

$$1 - \rho_{\max} \leq \frac{4}{\sqrt{\left( \arctan \frac{2}{\sqrt{5}} - \frac{2}{3} \right) \sqrt{5}}} \cdot \frac{1}{\sqrt{L}} \tag{4.2}$$

The compression of the preactivation range does not happen in chaotic initialization. As shown on figure 3b, the correlation of preactivations on any pair of different inputs approaches $\rho_{\mathrm{fp}} < 1$, i.e. the fixed point of the function $P$ defined in 2.3. With the approximation $\rho_{\max} \approx \rho_{\mathrm{fp}}$, theorem 1 predicts

$$\left\| \frac{\partial z^L}{\partial z^0} \right\|_{op} \geq \frac{\sqrt{\left( \arctan \frac{2}{\sqrt{5}} - \frac{2}{3} \right) \sqrt{5}}}{4} \cdot \left( 1 - \rho_{\mathrm{fp}} \right) \cdot \sqrt{L} \tag{4.3}$$

As shown in 3d, norm of the Jacobian indeed grows, causing large gradients in low layers (figure 3f).

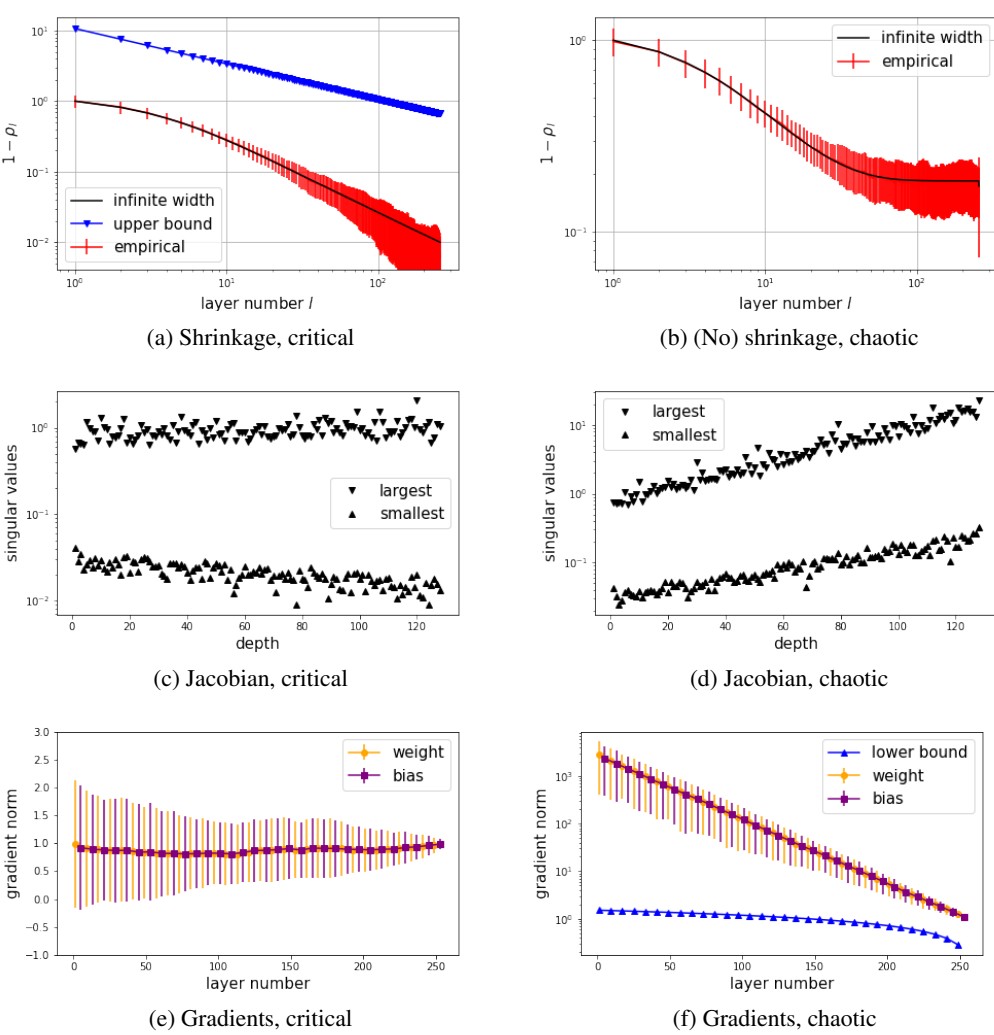

(a) Shrinkage, critical

(b) (No) shrinkage, chaotic

(c) Jacobian, critical

(d) Jacobian, chaotic

(e) Gradients, critical

(f) Gradients, chaotic

Figure 3: Representation shrinkage, singular values of the Jacobian and gradient propagation in deep erf networks. Bounds in figures 3a and 3f are the predictions of inequalities 4.2 and 4.3 respectively.

## 4.2 REMOVING REDUNDANT LAYERS

Here we compare the training of a full deep network, and a "skimmed" network obtained by replacing a block of middle layers with a single linear transformation. We hypothesize that since all inputs

to the middle block lie in a small region, it can be replaced with its linearization without significantly affecting the network outputs.

Figure 4 compares training of full and skimmed networks on MNIST. Full networks has 82 critically-initialized layers, alternating between ReLU and erf; "skimmed" one has 67 layers (a block of 16 is linearized). At initialization, preactivations entering the middle block have correlation at least $\approx 0.97$, on any possible pair of inputs; as shown on figure 4a, this does not change significantly after training. Figure 4c shows that their performance improves at similar rates, suggesting that the full and skimmed models follow similar trajectories. We show Pearson correlation between total displacements of the parameters in each layer on figure 4b. It is large (up to $0.8$) further from the linearized block, suggesting that these layers learn similar features; this is smaller ($\approx 0.1$) but positive in layers closer to the linearized block, suggesting some but not complete overlap.

## 5 DISCUSSION

### 5.1 THEORETICAL AND PRACTICAL IMPLICATIONS

We have demonstrated a problem with increasing the depth of infinitely wide fully-connected networks. It holds quite generally and appears to be a universal property of the fully-connected architecture rather than a particular class of activation functions or initialization schemes. It also demonstrates that critical initialization alone does not eliminate all the problems with signal propagation.

Our result demonstrates that vanilla MLPs are not the right model for networks of large-but-comparable depth and width. If $L$ and $N_l$ grow simultaneously, then the first block of $\log L$ layers is already problematic and the function $\rho_{\log L}(\rho_0)$ (as in figure 1) becomes pathological. We cannot avert it by using critical initialization or mixing different layer hyperparameters. This also hints that "weakly nonlinear networks" might make more sensible models in this setting.

Theorem 2 suggests that for ResNet initialization, breaking the symmetry between parameters is more important than specifying a good prior. From the theory side, we can deduce that well-behavedness of deep ResNets requires scaling the weight variance as $L^{-1}$, agreeing with Hayou et al. (2021). From the practical side, our result suggests choosing the variance below the threshold at which theorem 2 becomes equality for acceptable values of $\left\|\frac{\partial z^{(L)}}{\partial z^{(0)}}\right\|_{op}$ and $\rho_{\max}$.

In our derivation we assumed that all affine layers are followed by layer normalization. This simplifies the final result but does not restrict generality – in a network in which pointwise nonlinearity follows an affine map, we can insert layer normalization between them and rescale the activation function by the average norm of preactivations. This construction requires knowing preactivation norm in a network without layer normalization, which was studied e.g. in Poole et al. (2016).

One can note that $O(L^{-\frac{1}{2}})$ rate of decay of $1 - \rho_L$ in equations 4.1 and 4.2 is worse than asymptotic $O(L^{-2})$ for ReLU and $O(L^{-1})$ for erf. Moreover, the right-hand side of inequality 1 is additive and not multiplicative. This raises the suspicion that our result can be improved. We leave the investigation of sharper bounds for future work.

### 5.2 REMEDYING STRATEGIES

The proof of theorem 1 shows that to avoid representation shrinkage we somehow need to break the convexity of $P$-functions. Mixing activation functions cannot completely cancel out each other's undesirable behaviour, because each one only "adds convexity". It also highlights a problem common to layer normalization and self-normalizing networks – those methods only normalize the diagonal of the covariance matrix of a data batch. This ensures $O(1)$-scaling of preactivations, but off-diagonal behaviour may still cause representation shrinkage or gradient explosion. It was showed in Yang et al. (2019) that batch normalization is not a complete solution either – it necessarily leads to gradient explosion (which is slowest with linear activations).

Our result demonstrates limitations of normalization methods and critical initialization in improving the trainability of deep networks, and suggests ==using trainable nonlinearities, initialized as linear functions at the start of the training (similarly to the idea from== He et al. (2015)==).==

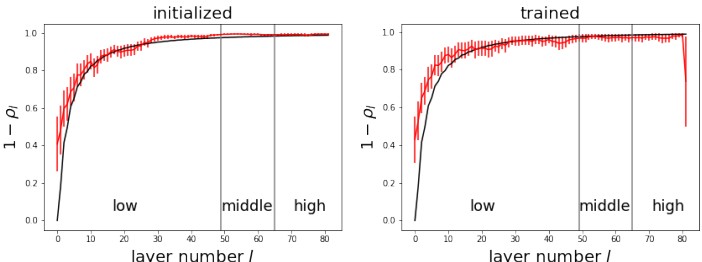

(a) Shrinkage in the full model before and after training. Empirical in red, infinite-width prediction in black

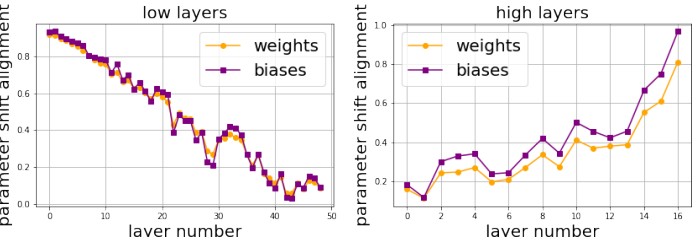

(b) Alignment of parameter displacement

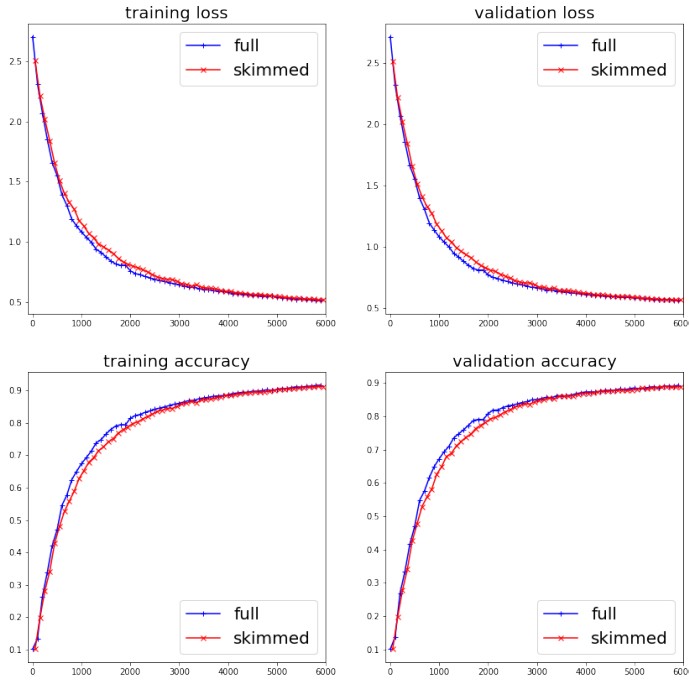

(c) Training curves

Figure 4: Full and "skimmed" network on MNIST

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

## A  PROOF DETAILS

### A.1  PROOF OF THEOREM 1

We will analyse the propagation of preactivations for two arbitrary data points $x_\alpha, x_\beta$. Following equation 2.2, denote the cosine similarity of preactivations at $l$-th layer as $\rho_l = \frac{z^{(l)}(x_\alpha) \cdot z^{(l)}(x_\beta)}{N_l}$. Let $P_l$ describe the effect of $l$-th layer on correlation, as defined in equation 2.3. Then for layers $k < l$ we have

$$\rho_l = P_l \circ P_{l-1} \circ \cdots \circ P_{k+2} \circ P_{k+1}(\rho_k)$$

Here we use a "physical" argument to relate $P_l$ with the Jacobian. For mathematically rigorous discussion see section A.1.1. If we let one input approach the other $x_\beta \to x_\alpha$, then

$$\limsup_{x_\beta \to x_\alpha} \frac{N_L^{-\frac{1}{2}} \|z^{(L)}(x_\alpha) - z^{(L)}(x_\beta)\|}{N_0^{-\frac{1}{2}} \|x_\alpha - x_\beta\|} = \left\| \frac{\partial z^{(L)}}{\partial z^{(0)}}(x_\alpha) \right\|_{op}$$

On the other hand, this scenario is equivalent to $\rho_0 \to 1$, and

$$\frac{N_L^{-1} \|z^{(L)}(x_\alpha) - z^{(L)}(x_\beta)\|^2}{N_0^{-1} \|x_\alpha - x_\beta\|^2} = \frac{2 - 2\rho_L}{2 - 2\rho_0} = \frac{1 - P_L \circ \cdots \circ P_1(\rho_0)}{1 - \rho_0} \xrightarrow{\rho_0 \to 1} \left( P_L \circ \cdots \circ P_1 \right)'(1)$$

Therefore the Jacobian has norm at least

$$\left\| \frac{\partial z^{(L)}}{\partial z^{(0)}} \right\|_{op}^2 \geq \left( P_L \circ \cdots \circ P_1 \right)'(1) \tag{A.1}$$

Maximal correlation can be expressed using $P_l$ as

$$\rho_{\max} = \max_{1 \leq l \leq L} \min_{\rho \in [-1,1]} P_l \circ \ldots P_1(\rho)$$

Let us replace this quantity with something more convenient. Define $\tilde{\rho}_l = P_l \circ \cdots \circ P_1(0)$ and $\tilde{\rho}_{\max} = \max_{1 \leq l \leq L} \tilde{\rho}_l$. By observation 1, $P_l$ are power series with all coefficients non-negative, so for $\rho \in [0, 1]$ we have

$$P_l \circ \cdots \circ P_1(\rho) \geq P_l \circ \cdots \circ P_1(0) = \tilde{\rho}_l$$
$$P_l \circ \cdots \circ P_1(-\rho) \geq 2 P_l \circ \cdots \circ P_1(0) - P_l \circ \cdots \circ P_0(\rho) \geq 2\tilde{\rho}_l - 1$$

Therefore $\rho_l \geq 2\tilde{\rho}_l - 1$, and $1 - \rho_{\max} \leq 2(1 - \tilde{\rho}_{\max})$.

If we restrict our attention to $\rho \in [0, 1]$, then $P_l \circ \cdots \circ P_1$ takes values in $[\tilde{\rho}_l, 1]$, so $P_{l+1}$ "encounters values only from this inverval". Therefore, let us "crop" all the $P$'s to functions $\tilde{P} : [0, 1] \circlearrowleft$ by defining

$$\tilde{P}_l(t) \overset{\text{def}}{=} \frac{P_l(\tilde{\rho}_{l-1} + (1 - \tilde{\rho}_{l-1})t) - \tilde{\rho}_l}{1 - \tilde{\rho}_l} \qquad \text{so that} \qquad P_l\left(\tilde{\rho}_{l-1} + (1 - \tilde{\rho}_{l-1})t\right) = \tilde{\rho}_l + (1 - \tilde{\rho}_l)\tilde{P}_l(t)$$

Then the $P_l, \tilde{P}_l$ are related by affine transforms, domain and codomain of $\tilde{P}_l$ is $[0, 1]$ and it satisfies $\tilde{P}_l(0) = 0, \tilde{P}_l(1) = 1$. Then we have

$$P_l \circ \cdots \circ P_{k+1}\left(\tilde{\rho}_k + (1 - \tilde{\rho}_k)t\right) = \tilde{\rho}_l + (1 - \tilde{\rho}_l) \cdot \tilde{P}_l \circ \cdots \circ \tilde{P}_{k+1}(t) \tag{A.2}$$

and as a consequence

$$\left( P_L \circ \cdots \circ P_1 \right)'(1) = (1 - \tilde{\rho}_L)\left( \tilde{P}_L \circ \cdots \circ \tilde{P}_1 \right)'(1) \tag{A.3}$$

When we substitute the formula 2.3, we see that $\tilde{P}_l$ is exactly a "cropped" Gaussian covariance of the activation function, i.e.

$$\tilde{P}_l(t) = \frac{\mathfrak{C}[\phi_l|\tilde{\rho}_{l-1}+(1-\tilde{\rho}_{l-1})t]-\mathfrak{C}[\phi_l|\tilde{\rho}_{l-1}]}{\mathfrak{C}[\phi_l|1]-\mathfrak{C}[\phi_l|\tilde{\rho}_{l-1}]}$$

We will now make use of the excess convexity $e$. It is a functional that takes convex functions on $[0,1]$ and returns real numbers. Its definition and all the necessary proofs are in appendix A.3. All we need to know in this proof is that it satisfies inequalities A.4, A.5 and A.6, which follow directly from its properties 4, 5 and 7.

The steepness estimate (property 4) states

$$\left(\tilde{P}_L \circ \cdots \circ \tilde{P}_1\right)'(1) \geq 1 + e\left(\tilde{P}_L \circ \cdots \circ \tilde{P}_1\right) \tag{A.4}$$

The superadditivity with respect to composition (property 5) states

$$e\left(\tilde{P}_L \circ \ldots \tilde{P}_1\right) \geq \sum_{l=1}^{L} e(\tilde{P}_l) \tag{A.5}$$

Finally, applying the "cropping" property 7 yields

$$e(\tilde{P}_l) \geq \frac{(1-\tilde{\rho}_{l-1})\|\phi_l-\overline{\phi}_l\|_{\mathcal{N}}^2}{2\|\phi_l'\|_{\mathcal{N}}^2} \tag{A.6}$$

This is all we need from $e$ to complete the proof. Combining inequalities A.4, A.5 and A.6 gives

$$\left(\tilde{P}_L \circ \cdots \circ \tilde{P}_1\right)'(1) \geq e\left(\tilde{P}_L \circ \cdots \circ \tilde{P}_1\right) \geq \frac{1-\tilde{\rho}_{\max}}{2} \sum_{l=1}^{L} \frac{\|\phi_l-\overline{\phi}_l\|_{\mathcal{N}}^2}{\|\phi_l'\|_{\mathcal{N}}^2}$$

Reexpressing LHS in terms of the Jacobian norm from equation A.1 and remembering $1 - \tilde{\rho}_{\max} \geq \frac{1-\rho_{\max}}{2}$ and equation A.3 we arrive at the desired inequality.

**Remark 2.** *So presented proof omits some details needed to construct the upper bound on $P_L \circ \cdots \circ P_1$ presented on figure 1 (blue line). This bound can be derived by combining equations A.5 and A.6, and unpacking the definition of $e(\tilde{P}_L \circ \cdots \circ \tilde{P}_1)$ as follows.*

*We have "cropped" the function $P_L \circ \cdots \circ P_1$ according to equation A.2*

$$P_L \circ \cdots \circ P_1(t) = \tilde{\rho}_L + (1 - \tilde{\rho}_L)\tilde{P}_L \circ \cdots \circ \tilde{P}_1(t)$$

*such that $\tilde{P}_L \circ \cdots \circ \tilde{P}_1(0) = 0$. Then it is enough to take care of $\tilde{P}_L \circ \cdots \circ \tilde{P}_1$. Equations A.5 and A.6 provide a bound on the excess convexity of $\tilde{P}_L \circ \cdots \circ \tilde{P}_1$*

$$e\left(\tilde{P}_L \circ \cdots \circ \tilde{P}_1\right) \geq \frac{1-\tilde{\rho}_{\max}}{2} \sum_{l=1}^{L} \frac{\|\phi_l-\overline{\phi}_l\|_{\mathcal{N}}^2}{\|\phi_l'\|_{\mathcal{N}}^2}$$

*Using the property 1 of excess convexity we can bound $\tilde{P}_L \circ \cdots \circ \tilde{P}_1$ from above by a piecewise linear function (this gives exactly the bound on figure 1). The proof of this property and an intuitive interpretation is presented in appendix A.3.*

**Remark 3.** *Having arrived at equation A.3, we could have rewritten it using chain rule and bound each $\tilde{P}_l'(1)$ straight away. There are two reasons for using the excess convexity. First, it describes $P_L \circ \cdots \circ P_1$ on the whole of $[0,1]$ and not only in the neighbourhood of 1. Knowing that $e(\tilde{P}_L \circ \cdots \circ \tilde{P}_1)$ is large allows us to produce the blue upper bound on figure 1, and thus explain why $\rho_L$ is nearly constant for large range of $\rho_0$, and rapidly jumps to 1 when the data points are very close. Second, it rules out the situation where the pathological behaviour of $P_L \circ \cdots \circ P_1$ would be restricted to some $(1-\epsilon, 1)$ with $\epsilon \to 0$ as $L$ grows.*

### A.1.1 Justification for equation A.1

To arrive at equation A.1 we the swapped the order of limits $\lim_{n_l \to \infty}$ and $\limsup_{x_\beta \to x_\alpha}$. Here we give a rigorous treatment without this exchange. This will require some understanding of the excess convexity $e$.

Denote $E = \frac{1 - \rho_{\max}}{4} \sum_{l=1}^{L} \frac{\|\phi_l - \bar{\phi}_l\|_{\mathcal{N}}^2}{\|\phi_l'\|_{\mathcal{N}}^2}$, and pick $0 < \tau \leq \frac{1}{2+E}$. Equations A.5 and A.6 together imply that $e(\tilde{P}_L \circ \cdots \circ \tilde{P}_1) \geq E$. From property 1 of excess convexity $e$ it follows that

$$\tilde{P}_L \circ \cdots \circ \tilde{P}_1(1 - \tau) \leq (1 + E)(1 - \tau) - E =$$
$$= 1 - (1 + E)\tau$$

plugging the relation between $\tilde{P}_l$ and $P_l$ from equation A.2 gives

$$P_L \circ \cdots \circ P_1(1 - \tau) \leq \tilde{\rho}_{\max} + (1 - \tilde{\rho}_{\max})(1 - (1 + E)\tau) =$$
$$= 1 - (1 - \tilde{\rho}_{\max})(1 + E)\tau \tag{A.7}$$

Now, take any $x_\alpha, x_\beta$ satisfying $\|x_\alpha\|^2 = \|x_\beta\|^2 = n_0$, $\frac{x_\alpha \cdot x_\beta}{n_0} = 1 - \tau$. Consider

$$\frac{n_L^{-1}\|z^{(L)}(x_\alpha) - z^{(L)}(x_\beta)\|^2}{n_0^{-1}\|x_\alpha - x_\beta\|^2} = \frac{1 - \frac{z^{(L)}(x_\alpha) \cdot z^{(L)}(x_\beta)}{n_L}}{1 - \frac{x_\alpha \cdot x_\beta}{n_0}}$$

As $n_1, \ldots, n_L \to \infty$, this converges to

$$\frac{1 - P_L \circ \cdots \circ P_1\left(\frac{x_\alpha \cdot x_\beta}{n_0}\right)}{1 - \frac{x_\alpha \cdot x_\beta}{n_0}} = \frac{1 - P_L \circ \cdots \circ P_1(1 - \tau)}{\tau}$$

Convergence in probability was proved in Hanin (2021), while almost sure convergence for sigmoidal[9] or ReLU activations can be deduced from Daniely et al. (2017). Remembering equation A.7, this quantity is at least

$$\frac{1 - P_L \circ \cdots \circ P_1(1 - \tau)}{\tau} \geq (1 - \tilde{\rho}_{\max})(1 + E)$$

Therefore as $n_1, \ldots, n_L \to \infty$ we have

$$\frac{n_L^{-1}\|z^{(L)}(x_\alpha) - z^{(L)}(x_\beta)\|^2}{n_0^{-1}\|x_\alpha - x_\beta\|^2} \geq (1 - \tilde{\rho}_{\max})E \tag{A.8}$$

By mean value theorem, there exists a point $x_\gamma$ on the line segment joining $x_\alpha, x_\beta$ satisfying

$$\left\|\frac{\partial z^{(L)}}{\partial z^{(0)}}(x_\gamma)\right\|_{op} \geq \frac{n_L^{-\frac{1}{2}}\|z^{(L)}(x_\alpha) - z^{(L)}(x_\beta)\|}{n_0^{-\frac{1}{2}}\|x_\alpha - x_\beta\|}$$

Combining this with the estimate A.8 yields

$$\left\|\frac{\partial z^{(L)}}{\partial z^{(0)}}(x_\gamma)\right\|_{op}^2 \geq \frac{1 - \rho_{\max}}{2} E = \frac{(1 - \rho_{\max})^2}{8} \sum_{l=1}^{L} \frac{\|\phi_l - \bar{\phi}_l\|_{\mathcal{N}}^2}{\|\phi_l'\|_{\mathcal{N}}^2} \tag{A.9}$$

Let us summarize the precise formulation. We have actually demonstrated two very similar statements that differ by the exact interpretation of convergence. Using the result of Hanin (2021), we obtain: on every line segment joining $x_\alpha, x_\beta$ (that satisfy $\|x_\alpha\|^2 = \|x_\beta\|^2 = n_0$ and $1 > \frac{x_\alpha \cdot x_\beta}{n_0} \geq \frac{1+E}{2+E}$), with probability tending to 1 as $n_1, \ldots, n_L \to \infty$, we can find a point $x_\gamma$ satisfying the inequality A.9. Employing the result of Daniely et al. (2017), the precise statement becomes: with sigmoidal[9] or ReLU activation, on every line segment joining $x_\alpha, x_\beta$ (that satisfy $\|x_\alpha\|^2 = \|x_\beta\|^2 = n_0$ and $1 > \frac{x_\alpha \cdot x_\beta}{n_0} \geq \frac{1+E}{2+E}$), almost surely, for sufficiently large $n_1, \ldots, n_L$, we can find a point $x_\gamma$ satisfying the inequality A.9.

---

[9]Satisfying $\|\phi_l\|_\infty, \|\phi_l'\|_\infty, \|\phi_l''\|_\infty < C\|\phi_l\|_{\mathcal{N}}$ for a (common) constant $C$

**Remark 4.** *The proof is simpler if in the statement of theorem 1 we replace the square of Jacobian norm on the LHS by its infinite-width analogue*

$$\limsup_{x_\beta \to x_\alpha} \frac{K_L(x_\alpha, x_\alpha) + K_L(x_\beta, x_\beta) - 2K_L(x_\alpha, x_\beta)}{K_0(x_\alpha, x_\alpha) + K_0(x_\beta, x_\beta) - 2K_0(x_\alpha, x_\beta)}$$

*where $K_l$ is the NNGP kernel* (de G. Matthews et al., 2018)

$$K_l(x_\alpha, x_\beta) = \lim_{n_1, \ldots, n_l \to \infty} \frac{1}{n_l} \sum_{i=1}^{n_l} z_i^{(l)}(x_\alpha) z_i^{(l)}(x_\beta)$$

### A.2   PROOF OF RESNET VARIANT 2

The proof follows the same strategy as in the fully-connected case: we write down $P_l$, "crop" it and recenter to $[0, 1]^2$, and apply the excess convexity functional.

We can work out the covariances

$$\frac{Y^{(l)}(x_\alpha).Y^{(l)}(x_\beta)}{\|Y^{(l)}(x_\alpha)\| \cdot \|Y^{(l)}(x_\beta)\|} = \rho_{l-1}$$

$$\frac{y^{(l)}(x_\alpha).y_i^{(l)}(x_\beta)}{n} = \rho_{l-1} + \sigma_{(l)}^2 \mathfrak{C}[\phi_l | \rho_{l-1}]$$

$$\rho_l = P_l(\rho_{l-1}) = \frac{z^{(l)}(x_\alpha).z^{(l)}(x_\beta)}{n} = \frac{\rho_{l-1} + \sigma_{(l)}^2 \mathfrak{C}[\phi_l | \rho_{l-1}]}{1 + \sigma_{(l)}^2 \|\phi_l\|_{\mathcal{N}}^2} \tag{A.10}$$

Where $P_l$ in equation A.10 again governs the change of correlation after $l$-th layer. Introduce $\tilde{\rho}_l = P_l \circ \cdots \circ P_1(0)$ and $\tilde{P}_l(t) = \frac{P_l(\tilde{\rho}_{l-1} + (1 - \tilde{\rho}_{l-1})t) - \tilde{\rho}_l}{1 - \tilde{\rho}_l}$. Similarly as in the fully-connected case, we have

$$\left\| \frac{\partial z^{(L)}}{\partial z^{(0)}} \right\|_{op}^2 = (1 - \tilde{\rho}_L)\big(\tilde{P}_L \circ \cdots \circ \tilde{P}_1\big)'(1) \tag{A.11}$$

Using properties 4 and 5 of excess convexity $e$ from appendix A.3 we can get the bound

$$\big(\tilde{P}_L \circ \cdots \circ \tilde{P}_1\big)'(1) \geq e\big(\tilde{P}_L \circ \cdots \circ \tilde{P}_1\big) \geq \sum_{l=1}^{L} e\big(\tilde{P}_l\big) \tag{A.12}$$

Now we only need a bound on $e(\tilde{P}_l)$. Substituting the equation A.10 to the definition of $\tilde{P}_l$ yields

$$\tilde{P}_l(t) = \frac{1 - \tilde{\rho}_{l-1}}{1 - \tilde{\rho}_{l-1} + \sigma_{(l)}^2 A} t + \frac{\sigma_{(l)}^2 A}{1 - \tilde{\rho}_{l-1} + \sigma_{(l)}^2 A} g(t)$$

where

$$A = \|\phi_l\|_{\mathcal{N}}^2 - \mathfrak{C}[\phi_l | \tilde{\rho}_{l-1}]$$

$$g(t) = \frac{\mathfrak{C}[\phi_l | \tilde{\rho}_{l-1} + (1 - \tilde{\rho}_{l-1})t] - \mathfrak{C}[\phi_l | \tilde{\rho}_{l-1}]}{A}$$

Now we bound the excess convexity of $\tilde{P}_l$. From property 8 we can deduce

$$e\big(\tilde{P}_l\big) \geq \frac{\sigma_{(l)}^2 A \cdot e(g)}{1 - \tilde{\rho}_{l-1} + \sigma_{(l)}^2 A}$$

And property 7 states

$$e(g) \geq \frac{(1 - \tilde{\rho}_{l-1})^2 \|\phi_l - \overline{\phi}_l\|_{\mathcal{N}}^2}{2A}$$

Which gives

$$e\big(\tilde{P}_l\big) \geq \frac{\sigma_{(l)}^2 (1 - \tilde{\rho}_{l-1})^2 \|\phi_l - \overline{\phi}_l\|_{\mathcal{N}}^2}{2\big(1 - \tilde{\rho}_{l-1} + \sigma_{(l)}^2 A\big)}$$

Finally we use $A \leq (1 - \tilde{\rho}_{l-1}) \|\phi_l'\|_{\mathcal{N}}^2$ to obtain

$$e\big(\tilde{P}_l\big) \geq \frac{\sigma_{(l)}^2 (1 - \tilde{\rho}_{l-1}) \|\phi_l - \overline{\phi}_l\|_{\mathcal{N}}^2}{2(1 + \sigma_{(l)}^2 \|\phi_l'\|_{\mathcal{N}}^2)}$$

Combining this with equations A.11, A.12 and $1 - \rho_{\max} \leq 2(1 - \tilde{\rho}_{\max})$ we get

$$\left\| \frac{\partial z^{(L)}}{\partial z^{(0)}} \right\|_{op}^2 \geq \frac{(1 - \rho_{\max})^2}{8} \sum_{l=1}^{L} \frac{\sigma_{(l)}^2 \|\phi_l - \overline{\phi}_l\|_{\mathcal{N}}^2}{1 + \sigma_{(l)}^2 \|\phi_l'\|_{\mathcal{N}}^2}$$

## A.3 EXCESS CONVEXITY

Here we define the excess convexity functional and prove its properties. We will be working with non-decreasing convex functions $g : [0, 1] \circlearrowleft$ satisfying $g(0) = 0, g(1) = 1$. Note that these conditions imply continuity.

Each $g$ satisfying these conditions must have an argument $t \in (0, 1)$ for which $g(1 - t) = t$. This follows from Darboux property of $g(1 - x) - x$; pictorially, the graph of $g$ must intersect the line $x + y = 1$. We then define the excess convexity of $g$ as

$$e(g) = \tfrac{1}{t} - 2$$

It turns out that despite a seemingly arbitrary definition, excess convexity possesses a number of nice properties:

**Property 1** (Bounding). *If $e(g) \geq C$, then $g(t) \leq \begin{cases} \frac{1}{1+C}t & \text{if} \quad t \leq 1 - \frac{1}{2+C} \\ (1 + C)t - C & \text{if} \quad t \geq 1 - \frac{1}{2+C} \end{cases}$*

*Proof.* The condition $e(g) \geq C$ means that the point $\frac{1}{2+C}\begin{pmatrix} 1 + C \\ 1 \end{pmatrix}$ lies above the graph of $g$.

Therefore the line segments joining it with $\underline{0}$ and $\begin{pmatrix} 1 \\ 1 \end{pmatrix}$ lie entirely above the graph of $g$. $\square$

**Property 2** (Non-negativity). *$e(g) \geq 0$, with equality only for $g = \mathrm{id}_{[0,1]}$*

*Proof.* By convexity $g\big(\tfrac{1}{2}\big) \leq \tfrac{1}{2}\big(g(0) + g(1)\big) = \tfrac{1}{2}$. Since $g(t) + t$ is increasing, we must have $t \leq \tfrac{1}{2}$ and hence $\tfrac{1}{t} - 2 \geq 0$. Strict inequality for $g \neq \mathrm{id}_{[0,1]}$ follows from property 3 below. $\square$

**Property 3** (Coerciveness). *We have the inequality $\|g - \mathrm{id}_{[0,1]}\|_\infty \leq \frac{e(g)}{1+e(g)}$*

*Proof.* Essentially, this proof boils down to the graph of $g$ lying in the shaded area.

If $x \geq 1 - t$ then convexity applied to arguments $0, 1 - t, x$ gives

$$t = g(1 - t) \leq \tfrac{x-1+t}{x}g(0) + \tfrac{1-t}{x}g(x) \qquad \Rightarrow \qquad g(x) \geq \tfrac{t}{1-t} \cdot x$$

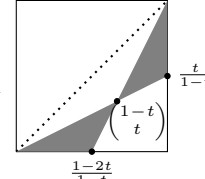

So $|x - g(x)| \leq 1 - \tfrac{t}{1-t} = \tfrac{1-2t}{1-t}$. Similarly, for $x \leq 1 - t$ we apply convexity to arguments $x, 1 - t, 1$ and obtain

$$x \geq g(x) \geq \max\left\{0, \tfrac{2t-1+(1-t)x}{t}\right\}$$

so again $|x - g(x)| \leq \tfrac{1-2t}{1-t}$ for $x \in [0, 1 - t]$. This means that

$$|x - g(x)| \leq \tfrac{1-2t}{1-t} = \tfrac{e(g)}{1+e(g)} \qquad \text{for all } x \in [0, 1]$$

$\square$

**Property 4** (Steepness estimate). *If $g$ is differentiable then $g'(1) \geq 1 + e(g)$*

*Proof.* By mean value theorem, there exists $s \in (1 - t, 1)$ satisfying

$$g'(s) = \frac{g(1) - g(1 - t)}{1 - (1 - t)} = \frac{1 - t}{t} = 1 + e(g)$$

Since $g$ is convex, its derivative is non-decreasing so $g'(1) \geq g'(s)$. $\square$

**Property 5** (Superadditivity with respect to composition). *For any two functions $g_1, g_2$ satisfying the conditions of this subsection we have $e(g_2 \circ g_1) \geq e(g_2) + e(g_1)$*

Let us first present the pictorial idea behind the proof. We know we can find $t_i$ with $g_i(1 - t_i) = t_i$. We examine the points

$$P_1 = \begin{pmatrix} 1 - t_1 \\ g_2(t_1) \end{pmatrix} \qquad P_2 = \begin{pmatrix} g_1^{-1}(1 - t_2) \\ t_2 \end{pmatrix}$$

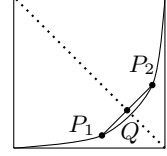

By construction, they lie on the graph of $g_2 \circ g_1$. We intersect the segment $P_1 P_2$ with the line $x + y = 1$ to get $Q$. Since $g_2 \circ g_1$ is convex, $Q$ lies above the graph of $g_2 \circ g_1$ and hence gives a lower bound on $e(g_2 \circ g_1)$.

*Proof.* Now we will make this hand-wavy proof rigorous. We start by a statement about the location of $P_1$. By convexity of $g_2$ we have

$$g_2 \circ g_1(1 - t_1) = g_2(t_1) \leq \tfrac{t_1}{1 - t_2} \cdot g_2(1 - t_2) + \tfrac{1 - t_1 - t_2}{1 - t_2} \cdot g_2(0) = \tfrac{t_1 t_2}{1 - t_2} \tag{A.13}$$

$t_i < \frac{1}{2}$, so the weights are non-negative. We can get an analogous inequality for $P_2$: using concavity of $g_1^{-1}$ we get

$$g_1^{-1} \circ g_2^{-1}(t_2) = g_1^{-1}(1 - t_2) \geq \tfrac{t_2}{1 - t_1} \cdot g_1^{-1}(t_1) + \tfrac{1 - t_1 - t_2}{1 - t_1} \cdot g_1^{-1}(1) = 1 - \tfrac{t_1 t_2}{1 - t_1}$$

$$g_2 \circ g_1 \left( 1 - \tfrac{t_1 t_2}{1 - t_1} \right) \leq t_2 \tag{A.14}$$

It is straightforward, if somewhat tedious, to check that the right (i.e. giving rise to something of the form $g_2 \circ g_1(1 - t) \leq t$) weights to combine inequalities A.13, A.14 are given by

$$w_1 = \tfrac{(1 - 2t_1)t_2(1 - t_2)}{(1 - t_1 - t_2)(t_1 + t_2 - 2t_1 t_2)} \qquad w_2 = \tfrac{t_1(1 - t_1)(1 - 2t_2)}{(1 - t_1 - t_2)(t_1 + t_2 - 2t_1 t_2)} \tag{A.15}$$

Adding inequalities A.13 with weight $w_1$ and A.14 with weight $w_2$ yields

$$g_2 \circ g_1 \left( 1 - w_1 t_1 - w_2 \tfrac{t_1 t_2}{1 - t_1} \right) \leq w_1 \cdot g_2 \circ g_1(1 - t_1) + w_2 \cdot g_2 \circ g_1 \left( 1 - \tfrac{t_1 t_2}{1 - t_1} \right) \leq w_1 \tfrac{t_1 t_2}{1 - t_2} + w_2 t_2$$

After substituting the weights from A.15 this simplifies to

$$g_2 \circ g_1 \left( 1 - \tfrac{t_1 t_2}{t_1 + t_2 - 2t_1 t_2} \right) \leq \tfrac{t_1 t_2}{t_1 + t_2 - 2t_1 t_2}$$

Therefore

$$e(g_2 \circ g_1) \geq \left( \tfrac{t_1 t_2}{t_1 + t_2 - 2t_1 t_2} \right)^{-1} - 2 = \tfrac{1}{t_1} + \tfrac{1}{t_2} - 4 = e(g_1) + e(g_2)$$

$\square$

**Property 6** (Cropping a power series). *Take a power series $g(x) = \sum_{k=0}^{\infty} c_k x^k$ with all coefficients non-negative $c_k \geq 0$. Let us "crop" it to the rectangle $[a, b] \times [g(a), g(b)]$ and "recentre"; formally, define*

$$\tilde{g}(s) = \tfrac{g(a + (b - a)s) - g(a)}{g(b) - g(a)} \qquad \text{so that} \qquad g(a + (b - a)s) = g(a) + \big(g(b) - g(a)\big) \cdot \tilde{g}(s)$$

*then $\tfrac{e(\tilde{g})(2 + e(\tilde{g}))}{1 + e(\tilde{g})} \geq \tfrac{(b - a)^2}{g(b) - g(a)} \sum_{k=2}^{\infty} c_k b^{k-2}$.*

*Proof.* Let $t = \tfrac{1}{2 + e(\tilde{g})}$ so that $\tilde{g}(1 - t) = t$. By the definition of $\tilde{g}$

$$tg(a) + (1 - t)g(b) - g\big(ta + (1 - t)b\big) = \big(g(b) - g(a)\big)\big(1 - t - \tilde{g}(1 - t)\big) =$$
$$= \big(g(b) - g(a)\big)(1 - 2t) =$$
$$= \big(g(b) - g(a)\big)\tfrac{e(\tilde{g})}{2 + e(\tilde{g})} \tag{A.16}$$

On the other hand, expanding $g$ yields

$$tg(a) + (1 - t)g(b) - g\big(ta + (1 - t)b\big) = \sum_{k=0}^{\infty} c_k \left[ ta^k + (1 - t)b^k - \big(ta + (1 - t)b\big)^k \right] =$$
$$= \sum_{k=2}^{\infty} c_k b^k \left[ t\big(\tfrac{a}{b}\big)^k + (1 - t) - \big(t \cdot \tfrac{a}{b} + 1 - t\big)^k \right] \tag{A.17}$$

where the constant $k = 0$ and linear $k = 1$ terms cancel. Now, consider the function $k \mapsto sx^k - (sx + 1 - s)^k$ for some $s, x \in [0, 1]$. We will prove that it is non-decreasing on $[1, \infty)$. Its derivative with respect to $k$ is

$$-sx^k \log \tfrac{1}{x} + (sx + 1 - s)^k \log \tfrac{1}{sx+1-s} \geq$$
$$\geq -sx(sx + 1 - s)^{k-1} \log \tfrac{1}{x} + (sx + 1 - s)^k \log \tfrac{1}{sx+1-s} =$$
$$= (sx + 1 - s)^{k-1} \Big[ s \cdot x \log x + (1 - s) \cdot 1 \log 1 - (sx + 1 - s) \log(sx + 1 - s) \Big]$$

which is non-negative by Jensen's inequality for $x \log x$. This allows to replace all brackets in A.17 with their values at $k = 2$, giving

$$\sum_{k=2}^{\infty} c_k b^k \Big[ t\big(\tfrac{a}{b}\big)^k + (1-t) - \big(t \cdot \tfrac{a}{b} + 1 - t\big)^k \Big] \geq \sum_{k=2}^{\infty} c_k b^k \Big[ t\big(\tfrac{a}{b}\big)^2 + (1-t) - \big(t \cdot \tfrac{a}{b} + 1 - t\big)^2 \Big] =$$
$$= t(1 - t)\big(1 - \tfrac{a}{b}\big)^2 \sum_{k=2}^{\infty} c_k b^k$$

Substituting $t = \frac{1}{2+e(\tilde{g})}$ and comparing to equation A.16 yields

$$\big(g(b) - g(a)\big) \tfrac{e(\tilde{g})}{2+e(\tilde{g})} \geq \tfrac{1+e(\tilde{g})}{(2+e(\tilde{g}))^2} (b - a)^2 \sum_{k=2}^{\infty} c_k b^{k-2}$$

which is equivalent to the desired inequality. $\square$

**Property 7** (Cropping a Gaussian covariance). *Suppose we "crop" a Gaussian covariance $\mathfrak{C}[f|\bullet]$ to $[r, 1]$ by defining*

$$\tilde{g}(t) = \tfrac{\mathfrak{C}[f|r+(1-r)t] - \mathfrak{C}[f|r]}{\mathfrak{C}[f|1] - \mathfrak{C}[f|r]}$$

*Then the excess convexity of $\tilde{g}$ is at least*

$$e(\tilde{g}) \geq \tfrac{(1-r)^2 \|f-\overline{f}\|_{\mathcal{N}}^2}{2\big(\|f\|_{\mathcal{N}}^2 - \mathfrak{C}[f|r]\big)} \geq \tfrac{(1-r) \|f-\overline{f}\|_{\mathcal{N}}^2}{2\|f'\|_{\mathcal{N}}^2}$$

*Proof.* Applying property 6 with $a = r, b = 1$ gives

$$2e(\tilde{g}) \geq \tfrac{e(\tilde{g})(2+e(\tilde{g}))}{1+e(\tilde{g})} \geq \tfrac{(1-r)^2}{\mathfrak{C}[f|1] - \mathfrak{C}[f|r]} \sum_{k=2}^{\infty} \hat{f}_k^2$$

Where $f = \sum_{k=0}^{\infty} \hat{f}_k h_k$ is the Hermite expansion of $f$. Notice that the sum is simply $\sum_{k=2}^{\infty} \hat{f}_k^2 = \|f - \overline{f}\|_{\mathcal{N}}^2$, which completes the proof of the first inequality.

To get the second one we only need to bound the denominator. Convexity of $\mathfrak{C}[f|\bullet]$ implies that

$$\mathfrak{C}[f|1] - \mathfrak{C}[f|r] \leq (1 - r) \cdot \tfrac{\partial \mathfrak{C}[f|\rho]}{\partial \rho}\Big|_{\rho=1}$$

From Stein's lemma we can deduce $\frac{\partial}{\partial \rho}\mathfrak{C}[f|\rho] = \mathfrak{C}[f'|\rho]$, so $\frac{\partial \mathfrak{C}[f|\rho]}{\partial \rho}\Big|_{\rho=1} = \|f'\|_{\mathcal{N}}^2$. $\square$

**Property 8** (Combining with identity). *For $p, q \geq 0, p + q = 1$ and a function $g$ we have*

$$e\big(p \cdot \mathrm{id}_{[0,1]} + q \cdot g\big) \geq \tfrac{qe(g)}{1+pe(g)}$$

*Proof.* Set $g(1 - s) = s, p(1 - t) + qg(1 - t) = t$. Write

$$g(1 - t) - t = \tfrac{p}{q}(2t - 1) \leq 0 = g(1 - s) - s$$

and since $g(1-x) - x$ is decreasing, we must have $1 - t \leq 1 - s$. This allows deduce from convexity of $g$ that $g(1 - t) \leq \frac{1-t}{1-s} g(1 - s) + \frac{t-s}{1-s} g(0) = \frac{s(1-t)}{1-s}$. Therefore

$$t - p + pt = qg(1 - t) \leq q \cdot \tfrac{s(1-t)}{1-s}$$

Substituting $\frac{s}{1-s} = \frac{1}{1+e(g)}$ and rearranging yields

$$t \leq \frac{1+pe(g)}{2+e(g)+pe(g)}$$

which is equivalent to

$$e\big(p \cdot \mathrm{id}_{[0,1]} + q \cdot g\big) = \frac{1}{t} - 2 \geq \frac{qe(g)}{1+pe(g)}$$

$\square$

## B  HERMITE EXPANSIONS OF COMMON ACTIVATION FUNCTIONS

Leaky rectified linear unit with negative slope $a$ is defined as $\mathrm{LReLU}_a(x) = ax + (1-a) \cdot \mathrm{ReLU}(x)$, and has the following

$$\mathfrak{C}[\mathrm{LReLU}_a|\rho] = a\rho + (1-a)^2 \frac{\sqrt{1-\rho^2} + \rho(\pi - \arccos \rho)}{2\pi}$$
$$\overline{\mathrm{LReLU}}(x) = \frac{1-a}{\sqrt{2\pi}} + \frac{1+a}{2}x$$

Therefore

$$\left\|\mathrm{LReLU}_a - \overline{\mathrm{LReLU}_a}\right\|_{\mathcal{N}}^2 = \frac{\pi-2}{4\pi}(1-a)^2$$
$$\left\|\mathrm{LReLU}_a'\right\|_{\mathcal{N}}^2 = \frac{1+a^2}{2}$$

And the critical initialization requires $\sigma_b = 0$.

The error function is defined as $\mathrm{erf}(x) = \frac{2}{\sqrt{\pi}} \int_0^x e^{-y^2} dy$. It induces

$$\mathfrak{C}[\mathrm{erf}|\rho] = \frac{2}{\pi} \arctan \frac{2\rho}{\sqrt{9-4\rho^2}}$$
$$\overline{\mathrm{erf}}(x) = \frac{2}{\sqrt{3\pi}}x$$

Therefore

$$\|\mathrm{erf} - \overline{\mathrm{erf}}\|_{\mathcal{N}}^2 = \frac{2}{\pi}\left(\arctan \frac{2}{\sqrt{5}} - \frac{2}{3}\right)$$
$$\|\mathrm{erf}'\|_{\mathcal{N}}^2 = \frac{4}{\pi\sqrt{5}}$$

And critical initialization happens when $\left(\frac{\sigma_b}{\sigma_w}\right)^2 = \frac{2}{\pi}\left(\frac{2}{\sqrt{5}} - \arctan \frac{2}{\sqrt{5}}\right)$.

