# OpenReview forum: "Gradient Explosion and Representation Shrinkage in Infinite Networks"
_ICLR.cc/2022/Conference — ICLR 2022 Submitted_

### Official Review · Reviewer_PiMb · 2021-10-24

**Correctness:** 3
**Technical Novelty And Significance:** 2
**Empirical Novelty And Significance:** 2
**Recommendation:** 5
**Confidence:** 3

**Main Review:**

The tradeoff between gradient explosion and representation shrinkage is an important question, and mean-field formalism is also a nice way to analyze this problem. But I found many parts of this paper very confusing. In particular, I have the following questions:

1. In Observation 1, what is the meaning of "existence and uniqueness of an attracting fixed point of $P$"? And why cannot $\rho$ be negative?
2. On the top of page 4, what's $P'(1)?$ In general, I think the three initialization regimes here require much more explanation.
3. In Theorem 1, how does the linearization error come into the analysis? Why does the tradeoff between gradient explosion and representation shrinkage disappear when the activation is linear?
4. On the top of page 5, why is it that "either the infimum of $P_L...P_1$ is close to 1 or its derivative at 1 is large"?
5. In Remark 1, what's $\kappa$ in $1-\rho_l = O(l^{-\kappa})$? And what's the precise definition of gradient explosion and vanishing here?
6. What does inequality 1 refer to in Section 4.1?
7. There is no explanation for Theorem 2. It might be good to compare Theorem 2 and Theorem 1, and also explain how Theorem 2 guides the choice of initialization variance in ResNet.
8. At the end of Section 5.2, what do you mean by "training should be started from a linear network"?


**Summary Of The Paper:**

1. This paper studied the signal propagation in a deep fully-connected neural network using the mean-field formalism, which assumes that the network width goes to infinity and the parameters are i.i.d. Gaussian. Theorem 1 proves that in a non-linear network, increasing the depth leads to either gradient explosion or representation shrinkage.
2. The theory is verified in the experiments, including the trade-off between non-linearity and representation shrinkage and the trade-off between gradient explosion and representation shrinkage.
3. On MNIST, it's shown that one can replace several non-linear middle layers of a deep network with a single linear transformation, and still gets a similar training trajectory.

**Summary Of The Review:**

I do think proving the tradeoff between gradient explosion and representation shrinkage is an interesting result. But this paper is not well written and many parts are very confusing to me (as pointed out in the main review). Currently, I think this paper is marginally below the acceptance threshold. I will consider raising my score if my questions are well addressed.

------------------------------------------------------------

Thanks for the response. After reading the response and other reviews, I decided to keep my original score.

---

> ### Author Response · Authors · 2021-11-15
> **Response to Reviewer PiMb**
>
> We thank the reviewer for their feedback. We have included the revisions (yellow). We will be grateful for pointing out any remaining shortcomings that do not meet ICLR standards.
>
> ---
>
> > 1.In Observation 1, what is "existence and uniqueness of an attracting fixed point of $P'(1)$"? Why can't $\rho$ be negative?
>
> **Meaning:** This refers to iterating $P$ on the interval $[-1,1]$. We claim there is exactly one fixed point (solution to$P(\rho_\text{fp})=\rho_\text{fp}$) that is attracting/stable (i.e. $\lim_{n\rightarrow\infty}P^n(\rho)=\rho_\text{fp}$ for all $\rho$ on some neighbourhood of $\rho_\text{fp}$, where $P^n$ is composition $P\circ\dots\circ P$).
>
> This matters when all layers have the same hyperparameters, so passing through many layers affects the correlation by repeated application of $P$. Then $\rho_l$ approaches a stable fixed point of $P$ as $l$ grows.
>
> **Pictorial pf:** $P$ is continuous and convex with $P(0)\geq 0$ and $P(1) = 1$. Thus, if we consider the graph of $P$ on $[0,1]$, it must intersect the line $y=x$ at exactly one point other than $x=1$, and have a slope smaller than 1 there; this is the fixed point $\rho_{fp}$. The only exception is when the whole graph of $P$ lies above $y=x$ and meets this line only at $x=1$, in which case $P'(1)\leq 1$, so $\rho=1$ is the (unique) stable fixed point. The first situation is the chaotic phase, while second entails critical ($P'(1)=1$) and ordered ($P'(1)<1$) regimes.
>
> The stable fixed point cannot be negative, because we just located it in $[0,1]$ (using continuity, convexity, $P(0)\geq 0,P(1)=1$).
>
> > 2.On page 4, what's $P'(1)$? I think the three regimes require more explanation.
>
> **Briefly:** $P'(1)$ is the derivative of $P$ at 1. $\rho=1$ is always a fixed point of $P$, and $P'(1)$ determines whether it is stable or unstable.
>
> We have $P(1)=1$, because every vector has correlation 1 with itself. For $\rho\approx 1$ we have $1-P(\rho)\approx P'(1)(1-\rho)$, so $1-P^k(\rho)\approx P'(1)^k(1-\rho)$. When $P'(1)<1$, applying $P$ moves the nearby points closer to 1; hidden representations of all data become similar. When $P'(1)>1$, iterating $P$ pushes $\rho$ away from 1; hidden representations of similar data become very different. This is an layer-normalized analogue of the well-known ordered/chaotic phase diagram (fig 1a in ["Deep Information Propagation"](https://arxiv.org/abs/1611.01232)).
>
> We have added more information about the regimes in the revision. Due to the space limitation (and the volume of literature on this topic), we decided to only state the main qualitative characteristics; however, we would be happy to incorporate any ideas improving the presentation.
>
> > 3.In Thm 1, how does the linearization error come into the analysis? Why does the tradeoff between gradient explosion and representation shrinkage disappear when activation is linear?
>
> **Idea:** With linear activation, correlation is preserved during forward propagation.
>
> **Why:** Zero linearization error implies a linear activation. Then $N_{l+1}^{-1}\mathbb{E}_{W^{(l+1)}}[z^{(l+1)}(x_\alpha)^\top z^{(l+1)}(x_\beta)]=N_l^{-1}z^{(l)}(x_\alpha)^\top z^{(l)}(x_\beta) $. This means that preactivation norms, covariances and correlations are perfectly preserved between layers. Thus $P_L\circ\dots\circ P_1$ is the identity, and $\rho_L$ as a function of $\rho_0$ exhibits no pathological behaviour.
>
> An intuitive reason for the appearance of linearization error is that it controls the "correlation distortion" by each layer.
>
> > 4.On page 5, why "either the infimum of $P_L\dots P_1$ is close to 1 or its derivative at 1 is large"?
>
> This statement is an intuitive explanation of a part of proof of theorem 1. This part roughly corresponds to equation A.2 from appendix A.1 (Proof of theorem 1).
>
> Informally, the steps are:
> - we write $P_L\dots P_1=a+(1-a)Q$ with $Q(0)=0,Q(1)=1$
> - we are able to show that $Q$ is "very convex" (think $y=x^\alpha$ on $[0,1]$ for larger and larger $\alpha$)
> - then $P_L\dots P_1$ is close to $a$ on $[0,1-\epsilon]$, and sharply increases to 1 on $[1-\epsilon,1]$, for some $\epsilon$
> - thus, one of the following holds:
>   - $a\approx 1$ and all values of $P_L\dots P_1$ close to 1, or
>   - $1-a$ is far from $0$ and $P_L\dots P_1$ has a large slope near 1.
>
> (we removed "either")
>
> ---
>
> 5) We replaced an unspecified constant $\kappa$ with explicit exponent $\frac{1}{2}$ that can be deduced from theorem 1 (assuming Jacobian norm and linearization errors are $O(1)$). Gradient explosion/vanishing is when the norm of gradients is exponential in $l$.
> 6) Inequality 4.1 is the prediction of theorem 1 applied to leaky ReLU networks. Similarly, 4.2 and the other inequality in subsection 4.1 are just theorem 1 applied to nets with erf activation.
> 7) We added an intuition behind a different behaviour of ResNets. We also included some implications in the discussion section.
> 8) We mean using trainable activativation functions (e.g. PReLU), set to linear at the start of training.

---

### Official Review · Reviewer_W3CL · 2021-10-27

**Correctness:** 2
**Technical Novelty And Significance:** 2
**Empirical Novelty And Significance:** 2
**Recommendation:** 3
**Confidence:** 3

**Main Review:**

This paper studies fully-connected networks in the infinite-width limit with layer norm applied at each layer before activations. To my knowledge this paper is first to study the signal propagation problem in networks with layer norm. Additionally, by decomposing activations in terms of an expansion of Hermite polynomials, the authors are able to consider the non-perturbative layer-to-layer evolution of the correlation of preactivations for networks consisting of nonlinear activation functions.

Unfortunately, I find the paper's notion of "representation shrinkage" to be not very intuitive, and I don't have a clear understanding of why it's not desirable. On page 4, the authors list a number of reasons for why this may be bad, but they do not provide strong evidence in support of these claims. Additionally, the authors say that it is a problematic that representation shrinkage occurs even at a polynomial rate (cf. "Remark 1"), but I would expect polynomial decay of correlations to be perfectly acceptable for the regimes of depth that are important in practice. Perhaps these claims would be stronger if they were connected -- either empirically or theoretically -- to desirable properties of trained networks, such as generalization error. Further to my point, as the authors discuss in their introduction, the infinite-width analysis will break down anyway for large enough depth. Thus, I think all these results should only be understood for intermediate regimes anyway.

I also think it would be helpful if the authors included in their high-level description of their results -- abstract, introduction, and one-sentence summary -- that they are focusing solely on fully-connected networks with layer norm.

Finally, the authors talk about gradient explosion throughout the paper, and heuristically, of course, there is a connection between preactivation correlation, input-output Jacobian, and gradients. However, I think it would be more clear to analyze the gradients directly, e.g. by studying gradient descent through the NTK formalism. With such an analysis, it becomes clear that there are a number of factors that can affect the gradient, and the preactivation correlations and input-output Jacobian are just part of the story.

One minor question: do we not have to be careful applying Theorem 1 to non-smooth activation functions such as ReLUs? For other analysis of such activation functions that I'm aware of, typically the computation of quantities detailing the correlation of different inputs requires a separate analysis.

One minor comment: the authors mention in their Discussion that their analysis supports searching for lighter models to promote green computing, but since the realistic regime that their analysis applies to already involves heavy models -- extremely wide networks such that the infinite-width approximation applies -- I don't see how this comment is really applicable.

**Summary Of The Paper:**

This paper studies signal propagation questions in wide fully-connected networks with layer norm. Its main result relates the correlation of preactivations as a function of layer to the input-output Jacobian of the network function, showing that the Jacobian is lower bounded by a function of the maximum correlation and the linearization error of activations. This analysis is then used to comment on the connection between the exploding gradient problem, choices of parameter initializations, and a phenomenon that the authors called "representation shrinkage."

**Summary Of The Review:**

I recommend rejecting this paper as it is current written. In particular, I think there needs to be more clarity around the authors new idea of "representation shrinkage" for the technical result to be appreciated. If possible, it would be nice if the claims about the importance of preventing this were further supported -- either theoretically or empirically -- by an analysis of its effect on training. I also think it would be better to focus directly on gradient updates to make claims about gradient explosion rather than using proxy quantities in terms of preactivation correlations and the input-output Jacobian. Finally, I would like to see some evidence of the paper's claim that critically initialized networks with polynomial decay of correlations are problematic in practice.

### After Author Responses
While I appreciate the response and comments of the authors, I stand by my original score.

---

> ### Author Response · Authors · 2021-11-20
> **Reply to Reviewer W3CL**
>
> We would like to thank the reviewer for their feedback. We included the revisions (green).
>
> ---
>
> ## Shrinkage
>
> **Intuition:** In some layer $l$ the hidden representations $z^{(l)}(x)$ of *all* data points $x$ are constrained to lie in a narrow region (the cone $\tfrac{v^\top z^{(l)}(x)}{n_l}>1-\epsilon$ for some $v$).
>
> **Practical significance:** For many practical applications polynomial decay is indeed acceptable, but we still think the community should be aware of this effect. However, a few works do report that making networks "more linear" at initialization alleviates training difficulties: ["Delving Deep into Rectifiers: Surpassing Human-Level Performance on ImageNet Classification"](https://openaccess.thecvf.com/content_iccv_2015/papers/He_Delving_Deep_into_ICCV_2015_paper.pdf) use parametric ReLU with negative slope closer to one, while ["ESRGAN: Enhanced Super-Resolution Generative Adversarial Networks"](https://arxiv.org/abs/1809.00219) (Supplementary section 2) decrease the variance of residual weights. We conjecture that the reason is representation shrinkage, but we did not investigate this due to space and time constraints.
>
> **Theoretical significance:** In the *intermediate regime* (with depth and width growing $n_l,L\rightarrow\infty$ but their ratio scaling as $\tfrac{depth}{width}=O(1)$) we can still look at the first $L_0$ layers, where $L_0$ is a large but fixed number, and demonstrate problems there. We believe that for theoretical study of models of feature learning, it will be valuable to know a huge limitation of one of the potential models.
>
> ---
>
> Regarding **analysing gradients directly**: There already is a body of works dealing with the theory of backpropagation directly (e.g. ["Deep Information Propagation"](https://arxiv.org/pdf/1611.01232.pdf)). However, all papers that we are aware of assume independence between weights and gradients (or that forward and backward propagation use independent parameters). While this gives predictions that agree with experiments, this assumption is strictly speaking not true, and we wanted to find a way of avoiding it.
>
> **Application to rough activations:** The proof works for any piecewise differentiable activation (that is square-integrable with respect to Gaussian weight), because the Hermite expansion is valid.

---

### Official Review · Reviewer_Vkqm · 2021-11-04

**Correctness:** 3
**Technical Novelty And Significance:** 3
**Empirical Novelty And Significance:** 3
**Recommendation:** 8
**Confidence:** 3

**Main Review:**

### Main Comments

* Beyond the mean field assumption, the bulk of the analysis in the paper is made without any additional assumptions, and applies to networks of any depth or activation function. Despite this generality, the insights derived from Theorem 1 and the subsequent experiments are quite useful, and represent fundamental tradeoffs inherent to the design of deep learning models.

* Although the main result is somewhat technical, the summarization of the proof method and subsequent takeaways of the result are done in a very reader-friendly manner. This makes the paper much more useful to a general audience than it would have been with a drier presentation.

### Minor Comments

* In Figures 2a and 3a, gradient shrinkage (as measured by 1 - \rho) relative to depth is bounded by a power-law relationship (blue curves), but in both cases the empirical behavior converges to a different power law with faster decay (black curves). The former prediction comes from Theorem 1, but could the authors elaborate on where this latter power law might come from?

* The black curves in Figure 2 are described as infinite depth in the figure caption, but infinite width in the figure legends.

**Summary Of The Paper:**

A mean field analysis of deep feedforward networks is given, which demonstrates a tradeoff between gradient stability, expressivity, and the choice of activation functions. This tradeoff is experimentally explored and verified for several network architectures, and high-level insights are given for designing better deep networks.

**Summary Of The Review:**

The paper gives a rigorous theoretical analysis that should be of interest to a wide swath of deep learning researchers and practitioners, and is presented in a very accessible manner. The experiments are well-chosen to complement this analysis.

---

> ### Author Response · Authors · 2021-11-15
> **Response to Reviewer Vkqm**
>
> We are very grateful for the feedback. We have fixed the typo in the caption of figure 2. We will welcome any other comments.
>
> ---
>
> Regarding the power laws, one can obtain better *asymptotic* exponents, as we demonstrate below. However, with this method we loose the non-perturbative character of the result, and only get information for "sufficiently large $l$" and "$\rho_l$ sufficiently close to 1". Because of this exponent discrepancy, we suspect that our result can be made sharper, but as of now we do not know any easy way to achieve this. A possible approach might be to use our method to find $l_0$ for which the correlation $\rho_{l_0}$ is in $(1-\epsilon, 1]$ for any pair of data points, and from that $l_0$ onwards employ perturbative analysis.
>
> Now, let us demonstrate the exponents we expect are optimal for $\rho_l\sim l^{-2}$ for ReLU and $\rho_l\sim l^{-1}$ for erf. The idea is straightforward - to analyse the iterates of $P$ using the Taylor expansion around fixed point $\rho=1$. The details go as follows:
>
> * For ReLU, one layer changes the correlation $\rho$ to $P(\rho)=\tfrac{\sqrt{1-\rho^2}+\rho(\pi -\arccos\rho)}{\pi}$. We can expand it near $\rho=1$ as $P(1-\epsilon)=1-\epsilon+\tfrac{2\sqrt{2}}{3\pi}\epsilon^\frac{3}{2}+O(\epsilon^2)$. Therefore correlations change as $\rho_{l+1} \approx \rho_l + \tfrac{2\sqrt{2}}{3\pi}(1-\rho_l)^\frac{3}{2}$. To extract the asymptotic behaviour we rewrite $\tfrac{1}{\sqrt{1-\rho_{l+1}}}\approx\tfrac{1}{\sqrt{1-\rho_l}}+\tfrac{\sqrt{2}}{3\pi}$. This leads to $\tfrac{1}{\sqrt{1-\rho_l}}\approx\tfrac{\sqrt{2}}{3\pi}l+const.$ and so $1-\rho_l\sim l^{-2}$.
>
> * For critically initialized erf networks, the next-layer correlation is $P(\rho)=1-\tfrac{\sqrt{5}}{2}\left(\arctan\tfrac{2}{\sqrt{5}}-\arctan\tfrac{2\rho}{\sqrt{9-4\rho^2}}\right)$. We can Taylor-expand it around $\rho=1$ and obtain $\rho_{l+1}\approx\rho_l+\tfrac{2}{5}(1-\rho_l)^2$. Again, we extract the asymptotics via $\tfrac{1}{1-\rho_{l+1}}\approx\tfrac{1}{1-\rho_l}+\tfrac{2}{5}$, but this time we get $1-\rho_l\sim\tfrac{5}{2l}$.

---

> > ### Comment · Reviewer_Vkqm · 2021-11-29
> > **Thank you for the explanation regarding scaling laws**
> >
> > Thank you for that clear derivation of the asymptotic exponents using Taylor expansions around the fixed-point behavior, that is very helpful!

---

### Official Review · Reviewer_vYPS · 2021-11-05

**Correctness:** 4
**Technical Novelty And Significance:** 2
**Empirical Novelty And Significance:** 3
**Recommendation:** 5
**Confidence:** 4

**Main Review:**

-- Strengths:
- The paper provides a general lower bound for norm of Jacobian of fully connected and residual networks. This bound can be used to deduce some implications for training of neural networks.

-- Main concerns:
- The authors only have included layer-wise normalizations in the definition of neural network architectures which seems to simplify the proofs, but it is not clear whether the results are applicable to commonly used networks that do not include such normalizations. The authors mention briefly that that the inclusion of such normalization does not affect the generality of the results, because the (average) normalizing factor can be factored in the nonlinearity, but I do not see how the result would change as this normalizing factor depends on the input and is not fixed for all the inputs. I would increase my score if the results are extended to a general fully connected network as it was not clear to me how it can be done without a fixed normalizing factor.

- In many of the plots, an upper bound is plotted in blue but there is no mention of what it is in the main body. I believe since it is used so often, there should be a mention of it or at least a clear pointer to the equation corresponding to this upper bound in the appendix.

- As the authors have mentioned in the literature review, there are many works that consider signal propagation through wide neural networks of different architectures in the past such as Lechao Xiao et al.
[Dynamical Isometry and a Mean Field Theory of CNNs: How to Train 10,000-Layer Vanilla Convolutional Neural Networks]
which consider convolutional networks and previous works that consider fully connected ones. At least for the case of convolutional networks it seems that very deep networks (e.g. with 10,000 layers) can be trained if initialized at the edge of chaos without any residual blocks or normalization. Is there something specific to convolutional architectures or does the same happen for fully connected networks as well? It seems like your theorems suggest that very deep fully connected networks should not be trainable but is 10,000 not deep enough?

- There are some works that derive a recursive form for computation of neural tangent kernel between two inputs for certain nonlinearities such as ReLU. See e.g. [On the Inductive Bias of Neural Tangent Kernels]. It seems to me that well-behavedness of NTK, for example if NTK is not exploding, should imply that the Jacobian is also well-behaved. It might be good to mention such works as well.

-- Minor comments/typos:
- Page 4, 2nd remark: $v$ is not defined in the definition of the cone

- Please add the assumption that $n_l\rightarrow \infty$ in Theorem 1.

- I believe that some of the effects described after the Theorem are not easy to understand the way they are written. The authors mention that at least one of these effects will necessarily occur. It would be helpful to add a description of when such effect might happen, for example in the second effect, one might say if the norm of the Jacobian is not large that necessarily means that $\rho_{max}$ is close to 1.


**Summary Of The Paper:**

The paper stablishes a lower bound for the Jacobian of a fully connected neural network based on the maximum correlation between layers and the error in linear approximation of nonlinearities with respect to Gaussian measure. A similar lower bound is derived for residual networks. The authors then mention some remarks about implications of this bound as well as verifying them through experiments.

**Summary Of The Review:**

The paper provides an interesting lower bound for the Jacobian of a neural network. But the main issue I have is that the neural network considered with normalization seems not to correspond to networks that are commonly used.

---

> ### Author Response · Authors · 2021-11-16
> **Response to Reviewer vYPS**
>
> We would like to thank the reviewer for providing valuable feedback. We incorporated the revisions (in blue).
>
> ---
>
> ## Getting rid of layer normalization
>
> **In short**: the factor does depend on the input, but only through its norm and deterministically. Therefore, we can restrict our attention to inputs of a fixed norm, and then apply our analysis.
>
> **To elaborate**: Consider an MLP (with the same notation as in the paper, but omitting layer norm), and denote $q_l=N_l^{-1}\lVert z^{(l)}\rVert^2$. As shown in ["Exponential expressivity in deep NN"](https://arxiv.org/abs/1606.05340), in wide limit it propagates as $q_l=\sigma_{(l),w}^2\mathbb{E}\Big[\phi_l\big(\mathcal{N}(0,q_{l-1})\big)^2\Big]+\sigma_{(l),b}^2$. In particular, if $x_\alpha,x_\beta$ have the same norm, then inductively $q_l(x_\alpha)=q_l(x_\beta)$, so we can use the same normalizing factor for them. Then adding layer normalization and replacing $\phi_l$ by $x\mapsto\phi_l(\sqrt{q_l}x)$ allows to deduce a problematic behaviour on the set of inputs of prescribed norm $q_0$.
>
> Furthermore, in "Exponential expressivity in deep NN" it was shwon that for a large class of activations $q_l\rightarrow q^*$ as $l$ grows, where $q^*$ depends only on the architecture and not on the input. Also it was observed that in practice $q^*$ is approached quite quickly. Thus in this case we can ignore first few layers, and our result will apply to the remaining block. Then there is no restriction on the input norm, because it gets uniformized by the first few ignored layers, so we actually can use a single normalizing factor.
>
> One qualitive difference is the possibility of $q_l\rightarrow 0$. Then our construction is "zooming in", and the rescaled activation approaches linear function of slope $\phi_l'(0)$. But in this case we run into another problem - the signals vanish during forward propagation. Linear functions are closed under scaling, so when $q^*=O(1)$, this construction cannot decrease the linearization error significantly (by more than a multiplicative constant).
>
> ---
>
> ## Convolutional networks
>
> We relied on the assumption of weight independence. Networks considered in "Dynamical Isometry and a Mean Field Theory of CNNs" do not satisfy these assumptions: first, there is weight sharing, and second, orthogonal initialization is employed. Our work was aimed at complementing the current understanding with a non-perturbative result in the simplest case; we think bridging it with more advanced work is an interesting direction, but probably outside the scope of the current paper.
>
> ---
>
> Regarding NTK behaviour: our work can be naturally expressed in the language of Neural Network Gaussian Process kernel. We agree that examining the NTK and RKHS would be a promising topic, but we think the overlap with this particular paper makes it more of a future direction rather than related work. It is possible that we missed some references though, and we would be grateful for recommendations of any papers that can place our work in a more complete context.

---

### Official Review · Reviewer_ZVUC · 2021-11-05

**Correctness:** 2
**Technical Novelty And Significance:** 2
**Empirical Novelty And Significance:** Not applicable
**Recommendation:** 5
**Confidence:** 4

**Main Review:**

Understanding the representation power, trainability of networks (e.g., vanishing/exploding gradients) is an important research direction in deep learning. Inspired by the "mean-field" approach (e.g., Poole), the authors study the lower bound of the operator norm of the input-output Jacobian map in the large width limit and discover a trade-off between representation power and trainability (exploding gradients). Although I find this line of research of great interest, the current paper has two main drawbacks that I couldn't recommend for acceptance.

1. Similar results have been considered in existing work, e.g. [1].

2. The proof of the main theorem is not correct (see details below).

In addition, it seems most of the observations from the current paper have been found in existing work (e.g., Schoenholz and follow-up work). The key difference is that the current work does not require the number of layers to go to infinity.

---

[Strength]: the paper is well-written, easy to follow, and with a very clear goal.

[Weakness]:
(1) There is a large gap in the proof of Theorem 1.
(2) Missing discussion of the line of research using random matrix theory to understand the input-output Jacobian [1], which also consider the operator norm of the input-out Jacobian and draws a very similar conclusion, e.g., the squared operator norm must grow linearly with the number of layers; see eq (17) and follow up discussion in [1].

---

In what follows, I elaborate (1) and (2) since they are related.

The biggest issue I see in the proof is the equation above (A.1) on page 11. The authors mixed the calculation of finite width networks (on the left of the equation) and infinite width network calculation together (on the right). More precisely, the authors exchange the order of the two limits $\lim_{width\to\infty} $ and $\limsup_{x_\alpha\to x_\beta}$. The exchangeability of the two limits is questionable to me. In the order: $\lim_{width\to\infty} \limsup_{x_\alpha\to_\beta}$, we need to handle a product of random matrices (if we compute the Jacobian). This is indeed a core contribution of [1], who uses free probability theory to compute the whole spectrum of the singular values of the Jacobian (assuming certain free independence of the matrices). If we swap the limits (we shouldn't do this without justification) to $\limsup_{x_\alpha\to x_\beta}\lim_{width\to\infty} $, the problem itself is reduced to computing the derivative of the composed correlation map, which is much simpler. I think these two limits are not unchangeable in general. E.g., using the order $\limsup_{x_\alpha\to x_\beta}\lim_{width\to\infty} $, both critical gaussian and orthogonal initialization give the same answer. But using the order $\lim_{width\to\infty}\limsup_{x_\alpha\to x_\beta} $, gaussian and orthogonal initialization can give different answers, see eq (17) vs (22) in [1].

---


Several Qs:

Q1:

How Theorem 1 leads to the four possible cases after it needs more discussion. In addition, what are the new insights quotient the existing ones from the order-chaotic analysis? It seems: the first case corresponds to the chaotic phase, the second case corresponds to the order phase. The third/fourth cases seem to be a finer analysis of the critical regime.

Q2: Remark1 the critical initialization. Several works have already identified the issue of the polynomial rate convergence of the correlation to 1 for Relu and smooth functions; see Proposition 1 in [2]; sec B.3. in [3].

Q3: I can't find places to explain the legends "upper bound",  "largest found".

Q4: How does Thm1 imply eq (4.1)? Do you assume the operator norm is bounded by O(1)?



[1] Resurrecting the sigmoid in deep learning through dynamical isometry: theory and practice,  https://arxiv.org/pdf/1711.04735.pdf
[2] On the Impact of the Activation Function on Deep Neural Networks Training, https://arxiv.org/abs/1902.06853
[3] Disentangling Trainability and Generalization in Deep Neural Networks, https://arxiv.org/abs/1912.13053

Minors comments:
1.) What is the domain of the inputs? It seems they are lying in the same sphere, not mentioned in the paper.






**Summary Of The Paper:**

The paper studies the operator norm of the input-output jacobian fully-connected networks with layernorm. Under the assumptions that the width approaches infinity, the authors provide a lower bound for the operator norm. Using this result, they argue that for very depth neural network, either this norm is very large or the network's representation power shrinks.

**Summary Of The Review:**

In sum, there seems to be a large gap in the proof of the main theorem. Even this gap has been fixed; it seems that a large portion of the paper's results overlaps with some existing work.

---

> ### Author Response · Authors · 2021-11-18
> **Reply to Reviewer ZVUC**
>
> First, huge thanks for getting to the internals of the proof! We filled the gap and mentioned RMT (updates in red).
>
> ---
>
> ## Proof correction
>
> **Main idea:** We pick a pair of inputs $x_\alpha,x_\beta$ (with $\lVert x \rVert^2 = n_0, x_\alpha^\top x_\beta=n_0(1-\tau)$), that will be fixed as the widths grow. The rest of the proof lower-bounds the limit of $\tfrac{\lVert z^{(L)}(x_\alpha)-z^{(L)}(x_\beta)\rVert}{\lVert x_\alpha-x_\beta\rVert}$ as widths tend to $\infty$. By multidimensional __mean value theorem__ we have $\tfrac{\lVert z(x_\alpha)-z(x_\beta)\rVert}{\lVert x_\alpha-x_\beta\rVert}\leq\sup\{\lVert(\text{Jac}z)(x)|x=tx_\alpha+(1-t)x_\beta\text{ for }t\in[0,1]\}$. Hence for sufficiently large widths, on the segment joining $x_\alpha,x_\beta$ there exist an argument with large norm.
>
> Note that we used the knowledge about behaviour of $P_L\circ\dots\circ P_1$ away from 1.
>
> **Why this does not contradict [1]:** Updated argument avoids taking $\limsup_{x_\beta\rightarrow x_\alpha}$. Actually $(P_L\circ\dots\circ P_1)'(1)$ is the mean square singular value, which is a lower bound for the squared operator norm of the Jacobian. Hence $\limsup_{x_\alpha\rightarrow x_\beta}\lim_{\text{width}\rightarrow\infty}$ is the same in Gaussian and orthogonal cases; while $\lim_{\text{width}\rightarrow\infty}\limsup_{x_\alpha\rightarrow x_\beta}$ compute $\lambda_\max^2$ (which is larger).
>
> In our proof we essentially say that maximal eigenvalue is at least the average. While mean eigenvalue is already well understood, in no earlier work have we seen it being related to the maximal correlation $\rho_\max=\max_{x_\alpha,x_\beta}\text{corr}(z^{(L)}(x_\alpha),z^{(L)}(x_\beta))$.
>
> ## Relation to RMT
>
> Both RMT methods and our result draw the same main conclusion: increasing the depth leads to problems, even at critical initialization$^1$. However, showcased problems are different. RMT results demonstrate widening of the spectrum of the Jacobian, which happens in an infinitesimal neighbourhood of a data point. The underlying phenomenon in our work is the pathological behaviour of correlation map $P_L\circ\dots\circ P_1$; this is non-local, and large Jacobian norm is only a symptom of one of the failure modes.
>
> We believe the main advantages of our result over the existing RMT theory are:
> - It characterizes the propagation of data points at non-infinitesimal distance
> - We use completely different method that avoids the assumptions of free independence of layer Jacobians and independence between weights and gradients
> - The role of deviation of activation from a linear function is apparent
> - Effect of mixing activations if clear
> - As mentioned, our result is non-asymptotic in $L$ (no $+O(1)$ or "for sufficiently large $L$")
> - The same inequality holds across all choices of $\sigma_w,\sigma_b$
>
> $^1$Note that the example keeping dynamical isometry in [1] requires changing initialization variances with depth, decreasing $q^*$. The norm of preactivations becomes smaller, so in practice we are "zooming in" the activation function and make it more linear, resembling the third case from our paper.
>
> ---
>
> Q1: The most important insight is the quantification of the role of non-linearity in the trade-off. We added conditions for the first two cases.
>
> Q2: Compared to those works, our result gives an explicit inequality and avoids formulations asymptotic in $L$. We mentioned them for additional context.
>
> Q3: "Upper bounds" are predictions of inequalities (4.1-3). "Largest found" refers to minimal $1-\rho_L$ (minimal output correlation) maximized over input pairs $x_\alpha,x_\beta$. It was found by gradient descent on $x_\alpha,x_\beta$. We added the descriptions in captions.
>
> Q4: We set the LHS in theorem 1 to one. In the proof, we actually have $(P_L\circ\dots\circ P_1)'(1)\geq RHS$ before we replace the derivative by $\lVert\text{Jac}\rVert_{op}^2$, and at criticality we have $(P_L\circ\dots\circ P_1)'(1)=1$.
>
> 1.) For convenience the first layer is linear and followed by layer normalization. This is equivalent to having inputs on a sphere.
>
> [1] Resurrecting the sigmoid in deep learning through dynamical isometry: theory and practice

---

> > ### Comment · Reviewer_ZVUC · 2021-11-20
> > **reply**
> >
> > A quick reply (when reading the corrected proof.)
> >
> > I don't think you can apply weak law of large number to obtain (A.8). Because the numerator is only the sum of conditional independent random variables. The entries in the penultimate layer are not independent. I think you may need to apply concentration inductively for every layer.
> >
> > In addition, it should be "strong" law of large number (concerning almost sure convergence) rather than the weak law, which is about weak convergence. Finally, there should be a tiny $-\epsilon$ on the right hand side if $n_i$ are finite.

---

> > > ### Author Response · Authors · 2021-11-20
> > > **correction**
> > >
> > > We think Lemma 2.4 from "Random Neural Networks in the Infinite Width Limit as Gaussian Processes" ([https://arxiv.org/abs/2107.01562](https://arxiv.org/abs/2107.01562)) can be applied here.
> > >
> > >
> > > Using the notation from the paper, we set $A=\\{\alpha,\beta\\}$, $f(z_{i,\alpha},z_{i,\beta})=z_{i,\alpha}\cdot z_{i,\beta}$, $\mathcal{O}^{(l)}=z^{(l)}(x_\alpha)\cdot z^{(l)}(x_\beta)=\tfrac{1}{n_l}\sum_{i=1}^{n_l}z^{(l)}_i(x_\alpha)z^{(l)}_i(x_\beta)$. The lemma then gives convergence of $\mathcal{O}^{(l)}$ in probability as $n_1,\dots,n_L\rightarrow\infty$.
> > >
> > >
> > > Also, we do not need to include $\epsilon$ - note that to arrive at (A.8) we drop $1-\tilde{\rho}_\max$ from the equation immediately above (1+E becomes just E). This is a positive number, and plays the role of $\epsilon$. (theorem 1 is vacuously true when $\tilde{\rho}_\max=1$)
> > >
> > >
> > > We will be happy to clarify any further issues.

---

> > > > ### Comment · Reviewer_ZVUC · 2021-11-22
> > > > **reply**
> > > >
> > > > Yeah. That lemma will give you the convergence in probability but not almost surely. Indeed, I think the result from Daniely et al gives an exponential tail bound which implies a.s. convergence.  Thanks for the comment on $\epsilon$.

---

> > > > > ### Author Response · Authors · 2021-11-22
> > > > > **reply**
> > > > >
> > > > > Thank you for the valuable discussion, and for mentioning the reference! We have included it in the paper.
> > > > >
> > > > > ---
> > > > >
> > > > > For other readers of this discussion, here we provide a sketch of the two justifications for inequality (A.8) from section (A.1.1).
> > > > >
> > > > > The probability space we mention in this answer represents the randomness in parameters. An outcome can be thought of as a particular initialization of weights, and event as the weights belonging to a given (measurable) set.
> > > > >
> > > > > Denote the event
> > > > >
> > > > > $A_n=\left\\{\left\lvert\tfrac{1-\frac{z^{(L)}(x_\alpha)^\top z^{(L)}(x_\beta)}{n_L}}{1-\frac{x_\alpha^\top x_\beta}{n_0}}-\tfrac{1-P_L\circ\dots\circ P_1(1-\tau)}{\tau}\right\rvert\leq 1-\tilde{\rho}_\max\text{ for the network of width }n\right\\}$
> > > > >
> > > > > Note that the inequality (A.8) (for the width $n$) holds for every outcome in $A_n$.
> > > > >
> > > > > **Argument based on** ["Random Neural Networks in the Infinite Width Limit as Gaussian Processes"](https://arxiv.org/abs/2107.01562):
> > > > >
> > > > > Lemma 2.4 gives convergence of $\tfrac{z^{(L)}(x_\alpha)^\top z^{(L)}(x_\beta)}{n_L}$ in probability. This means $\lim_{n\rightarrow\infty}\mathbb{P}(A_n)=1$. Therefore inequality (A.8) holds with probability tending to 1 as $n\rightarrow\infty$.
> > > > >
> > > > > **Argument based on** ["Toward Deeper Understanding of Neural Networks: The Power of Initialization and a Dual View on Expressivity](https://arxiv.org/abs/1602.05897):
> > > > >
> > > > > For sigmoidal or ReLU activations, theorems 2. and 3. give a bound $\mathbb{P}(A_n')\leq e^{-\lambda n}$ for some $\lambda>0$ (where $A'$ is the set complement in the probability space). Then $\mathbb{P}(\bigcup_{n>N}A_n')\leq\tfrac{e^{-\lambda N}}{1-e^{-\lambda}}$ which means $\mathbb{P}(\bigcap_{n>N}A_n)\geq 1-\tfrac{e^{-\lambda N}}{1-e^{-\lambda}}$. The event in which an outcome belongs to $A_n$ for all sufficiently large $n$ is precisely $\bigcup_{N=1}^\infty\bigcap_{n>N}A_n$. By continuity of probability is has $\mathbb{P}\left(\bigcup_{N=1}^\infty\bigcap_{n>N}A_n\right)=\lim_{N\rightarrow\infty}\mathbb{P}(\bigcap_{n>N}A_n)=1$. This means that almost surely (with probability one), inequality (A.8) holds for all sufficiently large $n$.

---

> ### Comment · Reviewer_ZVUC · 2021-11-27
> **Update of scores**
>
> In light of fixing the major issue I brought up, I raised my score to 5 and I appreciate several contributions of the paper, e.g. avoiding the assumptions of free independence in RMT. However, I also think the contribution is not significant enough for acceptance comparing to several other papers that I vote for weak acceptance.
>
>
> Minor comments.
>
> What does this exactly mean "Unlike RMT, our method can handle pairs of data points with non-infinitesimal differences". Can you elaborate on it?

---

> > ### Author Response · Authors · 2021-11-29
> > **clarification**
> >
> > We would like to thank the reviewer for their engagement, which notably improved the quality of our paper.
> >
> > By "handling data points with non-infinitesimal differences" we mean that we look at the simultaneous propagation starting from two *different* (i.e. with a finite difference) inputs $x_\alpha\neq x_\beta$. This is in contrast to RMT methods - they analyse the Jacobian, which is a local quantity (describes a *single* input and its infinitesimal neighbourhood). The underlying phenomenon of our result is actually the distortion of *finite* distances; we chose to interpret one of the "failure modes" as a Jacobian blow-up because of its established role in the literature.

---

### Decision · Program_Chairs · 2022-01-20

**Decision:**

Reject

**Comment:**

This work performs a mean field analysis of a certain class of fully connected networks with and without layer normalization. Theory is provided which successfully predicts when some networks will exhibit either exploding gradients, or "representation shrinkage" which is similar to the extreme ordered phase discussed in prior works on signal propagation. The primary concerns raised by reviewers included, large overlap with prior works on signal propagation, a bug in the proof of the main theorem, lack of clarity, and many assumptions made in the theory which significantly limit the space of architectures for which the theory can be applied. Some of these concerns were addressed in the rebuttal period, notably major flaw in the main theorem was resolved and some concerns on clarity were addressed. However, with the remaining issues (notably overlap with prior work, and overly restrictive assumptions made) a majority of reviewers did not recommend acceptance in the end. The AC agrees with this final decision and recommends the authors look to further expand upon the contributions relative to prior work.